

# Global relevance of marine organic aerosols as ice nucleating particles

Wan Ting Katty Huang[1], Luisa Ickes[2,*], Ina Tegen[3], Matteo Rinaldi[4], Darius Ceburnis[5], and
Ulrike Lohmann[1]

[1]Institute for Atmospheric and Climate Science, ETH Zurich, Zurich, Switzerland
[2]Institute of Meteorology and Climate Research, Karlsruhe Institute of Technology, Karlsruhe, Germany
[3]Leibniz Institute for Tropospheric Research (TROPOS), Leipzig, Germany
[4]Institute of Atmospheric Sciences and Climate, National Research Council, Bologna, Italy
[5]School of Physics and Centre for Climate and Air Pollution Studies, Ryan Institute, National University of Ireland Galway,
Galway, Ireland
[*]now at: Department of Meteorology, Stockholm University, Sweden

*Correspondence to:* Katty Huang (katty.huang@env.ethz.ch)

**Abstract.**

Ice nucleating particles (INPs) lower the supersaturation required and/or increase the temperature at which supercooled droplets start to freeze. They are therefore of particular interest in mixed-phase temperature regimes, where supercooled liquid droplets can persist for extended periods of time in the absence of INPs. When INPs are introduced to such an environment,
the cloud can quickly glaciate following the Wegener-Bergeron-Findeisen process and possibly precipitate out, altering its radiative properties.

Despite their potential influence on climate, the ice nucleation ability and importance of different aerosol species is still not well understood and is a field of active research. In this study we used the aerosol-climate model ECHAM-HAM to examine the global relevance of marine organic aerosols (MOA), which have drawn much interest in recent years as a potentially important
INP in remote marine regions. We address the uncertainties in emission and ice nucleation activity of MOA with a range of reasonable set-ups and find a wide range of resulting MOA burdens. The relative importance of MOA as an INP compared to dust is investigated and found to depend strongly on the type of ice nucleation parametrisation scheme chosen. Regardless, MOA was not found to affect the microphysical properties of clouds or the radiative balance significantly, due to its relatively weak ice activity and a low sensitivity of cloud ice properties to heterogeneous ice nucleation in our model.

## 1 Introduction

In regions with scarce ice nucleating particles (INPs), liquid cloud droplets can remain supercooled for extended periods of time before the drops freeze homogeneously. In the presence of INPs, phase change is facilitated and supercooled cloud droplets can freeze at temperatures warmer than the homogeneous freezing temperature. Together with the Wegener-Bergeron-Findeisen process through which ice crystals grow at the expense of liquid droplets due to their difference in saturation vapour pressure,
INPs can alter the radiative properties of clouds and thus climate through glaciation and possible precipitation (Lohmann,





2002). Representation of INPs and their freezing ability is therefore of importance in climate modelling, especially for studies investigating aerosol-cloud effects.

Indeed, the subject of INPs is an area of active research in both modelling as well as laboratory and field work (e.g. Hoose and Möhler, 2012; Cziczo et al., 2017). Suggested INP candidates, such as mineral dust, terrestrial biogenic material, and black

carbon, are mostly of terrestrial origin. Recently, however, more interest has been drawn to oceans being possible sources of ice-active organic matter (Bigg, 1973; Knopf et al., 2011; Wang et al., 2015; Wilson et al., 2015; DeMott et al., 2016; McCluskey et al., 2017). While likely not as effective an INP as mineral dust, the difference in geographical locations of their emission sources may cause such marine organic aerosols (MOA) to become an important source of INPs in remote marine regions.

Marine organic aerosols can either be emitted directly as primary aerosol from the ocean surface by bubble bursting or

formed through a secondary process involving the condensation of biogenic volatile organic compounds (BVOCs) emitted from the ocean, and the resulting aerosols can either be water insoluble (WIOM; water insoluble organic matter) or water soluble (WSOM). The type relevant for ice nucleation is the insoluble organic matter, which forms mainly from primary emissions (Ceburnis et al., 2008). In this study, therefore, we will only focus on the primary emitted WIOM, and thus only refer to such WIOM when discussing MOA.

In investigating the global impact of MOA as INPs using general circulation models, an earlier study by Yun and Penner (2013) found MOA to be the dominant source of heterogeneously frozen ice crystals in the Southern Hemisphere compared to contributions from dust and black carbon. They also noted a better comparison of modelled ice water path to satellite observations from ISCCP when MOA are added as an additional source of INPs. Due to the lack of more measurement data at the time of publication, however, the representation of MOA ice-activity in their study is simply constrained by a fixed ratio of

nucleation efficiency at -15 °C that is 3 times higher over the Antarctic Ocean at 40° S than over Australia, based on Schnell and Vali (1976)'s evaluation of the Bigg (1973) INP data. It assumes, therefore, implicitly that MOA alone accounts for any shortfall in the model in representing the difference in ice nucleation ability of aerosols in Australia and over the Southern Ocean. This would render the calculated MOA ice-activity dependent on aerosol transport and the ice nucleation ability of other species in the model, while at the same time discounting other sources of INPs not yet considered. Thus while a better

agreement with observational data could be obtained, the attribution of MOA being the missing ingredient in climate models may not be entirely robust.

In terms of the relative contribution of MOA to the global INP population when compared to other sources, Burrows et al. (2013) found a greater contribution of MOA compared to terrestrial biogenic aerosols over nearly all regions except central continental areas, and a greater contribution compared to dust over the Southern Ocean. A recent paper by Vergara-Temprado

et al. (2017) also found MOA to be the dominant source of INP in remote locations, particularly in the southern high latitudes during Austral autumn to spring. Notably, they also found MOA to be the more dominant source of INP compared to dust (K-feldspar) on 10-30 % of days in the Northern Hemisphere.

MOA can also have impacts on climate through cloud properties of warm liquid clouds. This was investigated by Meskhidze et al. (2011) and Gantt et al. (2012), who concluded a weak influence of MOA on the global CCN concentration but up to a

20 % localized increase in the annually averaged low-level cloud droplet number concentration (CDNC) over remote oceans,



as well as up to a 7 % decrease in the anthropogenic aerosol indirect forcing (though their MOA emission rates remain fixed between present day and pre-industrial periods). These potential effects, however, will not be the focus of the current study.

The goal of this study is to quantify possible contributions of MOA to heterogeneous ice nucleation and their subsequent influence on cloud properties on the global scale. We hypothesize a potential impact in remote marine regions, and test our hypothesis while considering various uncertain aspects in the representation of MOA ice nucleation in a global climate model.

## 2 Methodology

### 2.1 The aerosol-climate model

Simulations in this study are performed using the aerosol-climate model ECHAM6.3-HAM2.3. The main atmospheric component is ECHAM6.3 (Stevens et al., 2013), except for a two-moment cloud microphysics scheme that is coupled to the aerosol module HAM (Lohmann et al., 2007). Aerosols are represented as a superposition of seven lognormal size distributions, representing aerosol populations in four size modes and two different mixing states, except for the nucleation mode (number median radius $\bar{r} \leq 0.005$ $\mu$m ) which only contains sulphate aerosols in the internally mixed/soluble mode. All other size modes (Aitken: $0.005$ $\mu$m $< \bar{r} \leq 0.05$ $\mu$m, accumulation: $0.05$ $\mu$m $< \bar{r} \leq 0.5$ $\mu$m, coarse: $0.5$ $\mu$m $< \bar{r}$) are divided into an internally mixed/soluble mode in which particles are assumed to contain a fraction of all species present, in particular the soluble sulphate aerosol, and an externally mixed/insoluble mode in which each particle is assumed to contain one species only. Only one size distribution (with one total number concentration, median radius, and standard deviation) is considered per mode, while the contribution of each species is represented by their individual masses, which are traced separately.

Various aerosol processes are explicitly represented as described in Zhang et al. (2012). Changes in recent model updates include the use of the Abdul-Razzak and Ghan (2000) scheme for activation of aerosols to form cloud droplets, which is based on Köhler theory, and the use of the Long et al. (2011) sea salt emission parametrisation with a sea-surface temperature dependence applied following Sofiev et al. (2011). Also, anthropogenic emissions are fixed at year 2000 levels in the following simulations and the minimum cloud droplet number concentration (CDNC) is 10 cm$^{-3}$. In the base version used in the current study, aerosol species considered include sulphate, dust, black carbon, organic carbon, and sea salt, among which dust is allowed to nucleate ice through immersion freezing, following either Ickes et al. (2017) or Niemand et al. (2012) as opposed to Lohmann and Diehl (2006) in the default model set-up. No other types of heterogeneous ice nucleation are considered. In the current study, MOA is implemented as an additional species in the internally mixed accumulation and coarse modes, as shown in Table 1 which lists the species present in each of the seven aerosol modes. Aitken mode MOA is not considered as our model does not consider sea spray production in that size mode. Additionally, MOA is allowed to nucleate ice through immersion freezing, as described in the following section.



**Table 1.** List of aerosol species present in each of the seven modes. In bold are tracers added in the current study.

| Size mode | Internally mixed/soluble | Externally mixed/insoluble |
|---|---|---|
| Nucleation | Sulphate | |
| Aitken | Sulphate, organic carbon (OC), black carbon (BC) | OC, BC |
| Accumulation | Sulphate, OC, BC, sea salt (SS), dust, **MOA** | Dust |
| Coarse | Sulphate, OC, BC, SS, dust, **MOA** | Dust |

## 2.2 MOA implementation

### 2.2.1 Emission of MOA

MOA emission is calculated online and dependent on the sea salt (SS) emission, such that the total sea spray emitted is the sum of the two (i.e. sea spray = SS + MOA), with the organic mass fraction (OMF) defined as $\mathrm{OMF} = \frac{\mathrm{MOA}}{\mathrm{total\ sea\ spray}} = \frac{\mathrm{MOA}}{\mathrm{MOA+SS}}$.

Sea salt is emitted following Long et al. (2011) and remains independent of the MOA emission for most cases. MOA is then emitted additionally as

$$\mathrm{MOA_{mass\ flux}} = \frac{\mathrm{SS_{mass\ flux}} \times \mathrm{OMF}}{1 - \mathrm{OMF}}. \tag{1}$$

The only exception is when MOA is emitted also following Long et al. (2011), in which case the sea salt emission is reduced due to partitioning of some of the emitted mass into MOA. The density of MOA is set to be 1000 kg m$^{-3}$ (Vignati et al., 2010), with radiative properties identical to that of organic carbon and a hygroscopicity parameter $\kappa$ of zero. Due to the lack of measurement data, the latter two properties are chosen for simplicity and, for the last case, consistency with other potential INP candidates. While sensitivity to the chosen radiative properties of MOA have yet to be investigated, a previous study by Gantt et al. (2012) have not shown a strong dependence of the results on the chosen hygroscopicity parameter.

No additional number flux due to MOA is considered, as we assume it to be always internally mixed with sea salt during emission. This is treated differently in different studies, with most emitting MOA as an internal mixture with sea salt (e.g. Long et al., 2011; Vergara-Temprado et al., 2017) while studies by Meskhidze et al. (2011) and Gantt et al. (2012) have noted a stronger impact of MOA on the modelled CDNC when they are assumed to be externally mixed during emission (that is, with an additional number flux but still emitted into internally mixed modes). Unfortunately, no measurement data is available to quantify such potential externally mixed number flux nor the division between internally and externally mixed emissions. Should some of the MOA be emitted separately from sea salt, they would be part of the externally mixed/insoluble mode in our model. However, as we are interested in the immersion freezing property of MOA, which require immersion of the aerosol in a cloud droplet that can only occur for soluble/internally mixed particles, for simplicity, we emit all of the MOA into the internally mixed mode directly in order to give an upper estimate of the potential impact of MOA as INP.

Various OMF parametrisations are available in the literature (e.g. Vignati et al., 2010; Gantt et al., 2011; Rinaldi et al., 2013; Burrows et al., 2014; Vergara-Temprado et al., 2017), which produce a wide range of MOA fluxes when applied to the global





scale, as was also shown in Meskhidze et al. (2011) and Lapina et al. (2011). A measure of marine biological activity is often used in these parametrisations, while some also consider a negative dependence on the near surface wind speed based on the argument of oceanic mixing leading to a reduction in surface organic enrichment. The performance of each parametrisation is thus also highly dependent on the model wind speeds and choice of representation for marine biological activity, in addition to

the model's sea salt emission.

In this study, only ocean surface chlorophyll is used to represent the marine biological activity. Despite ongoing debate on the validity of chlorophyll as a proxy for the organic fraction in emitted sea spray, it has been shown that there is currently no better alternative which has a global coverage of available data (Rinaldi et al., 2013; O'Dowd et al., 2015). Instead, we address the dependence on ocean biological activity data by using two different sources of chlorophyll datasets. In most simulations,

multi-year monthly mean Level 3 observational data from the Sea-viewing Wide Field-of-view Sensor (SeaWiFS; Hu et al., 2012) was fed into the model. Free simulations, as will be described later in Sect. 2.3, uses the full 12 years of available observational data from 1998 to 2010, while for the nudged simulations, a subset corresponding to the nudged period from March 2003 to May 2009 is used. The two choices of averaging time periods resulted in only very slight, localized differences in the chlorophyll concentrations (not shown). Such satellite-based observations, however, have a limited coverage in the

polar regions in the winter hemisphere that can create a data void as far equatorward as 50° (though in the less biologically active winter hemisphere). Also in light of the possibility to accommodate pre-industrial and future simulations, a sensitivity study is performed using chlorophyll concentration data from the CMIP5 multi-model ensemble outputs (Taylor et al., 2012). Monthly mean chlorophyll maps were created using results from the last six years (2000 to 2005) of the earth system model (ESM) historical simulations, from which only eight models contain chlorophyll data, as listed in Table A1 in the appendix.

Comparison of the two sets of maps is shown in Fig. 1. Notable deviations of the modelled data from observational means include the lack of peak values near coastlines, which could be due to unresolved coastal processes and averaging across models and/or errors in observations near coastlines, a more widespread coverage of medium concentrations in the high latitude regions of the spring-summer hemisphere, especially over the Southern Ocean, and a persistent local peak concentration in the equatorial upwelling region off the west coast of South America. The impact of such differences is discussed in the results.

Offline calculations were performed to compare the various OMF parametrisations when applied to our model to long-term observations at Amsterdam Island in the southern Indian Ocean (Sciare et al., 2009) and Mace Head in Ireland (Rinaldi et al., 2013), as described in Appendix A. The Rinaldi et al. (2013) parametrisation, which has a maximum OMF set to 78 %, was found to outperform others at both stations when applied to our model. Thus despite the circular logic of the parametrisation having been derived by using the exact same Mace Head data which we used for validation, the Rinaldi et al.

(2013) parametrisation is chosen for our control set-up. This does not, however, guarantee the most realistic emission rate when applied at the global scale. More long-term measurements from different parts of the globe would be required for a better validation of the model simulations.

MOA is emitted into the internally mixed accumulation mode and allowed to grow through coagulation and condensation of sulphate into the coarse mode. This is consistent with the Rinaldi et al. (2013) OMF parametrisation, which is based on

observations of submicron emissions. Previous studies have also noted a difference in the organic fraction of accumulation and



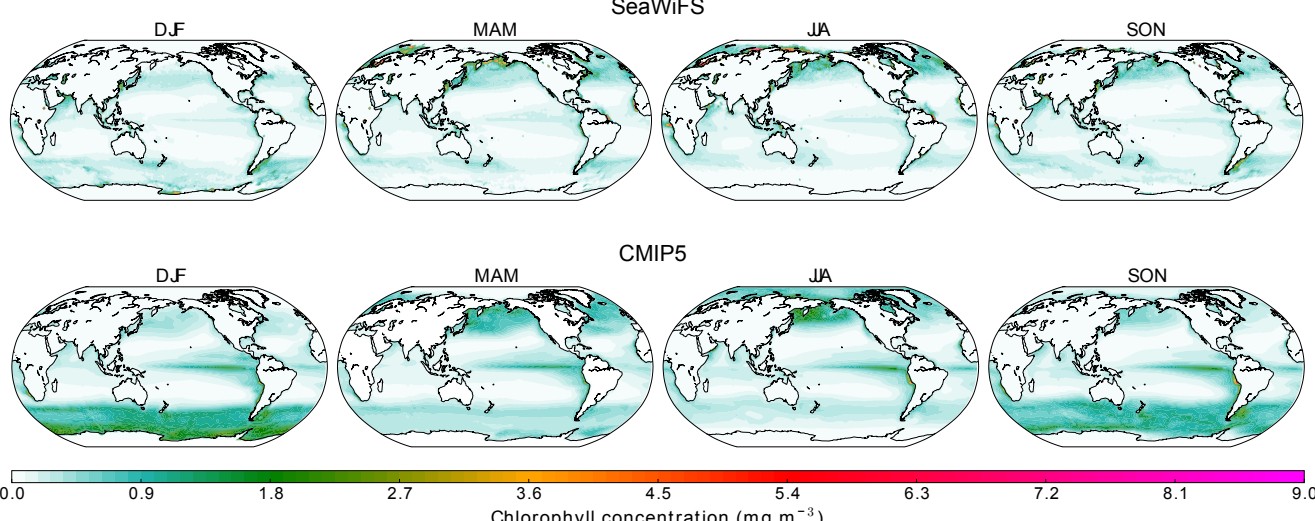

**Figure 1.** Maps of seasonal mean chlorophyll concentrations used as input files for the nudged simulations. The top row are maps from the SeaWiFS satellite observational dataset from March 2003 to May 2009, and the bottom row the mean from CMIP5 historical simulations for the years 2000 to 2005.

coarse mode sea spray, with higher fractions in the smaller size mode (Facchini et al., 2008). Thus it would not be appropriate to extrapolate the emission parametrisation to coarse mode particles and emission of MOA in the coarse mode is not considered in our simulations.

### 2.2.2 Heterogeneous ice nucleation of MOA

Quantification of the ice nucleation ability of MOA is still a topic of active research. Currently, only one published parametrisation is available in the literature, namely that of MOA immersion freezing from Wilson et al. (2015). This is an empirical fit to droplet freezing measurements performed using metal mesh and glass plate samples collected from the marine microlayer, which gives a purely temperature-dependent parametrisation for the number of INPs per mass of total organic carbon. It should be noted, however, that this parametrisation is developed based on sea surface microlayer samples, which does not necessary

reflect the concentration of INPs in the MOA that actually gets aerosolised and emitted into the atmosphere (McCluskey et al., 2017). To convert from the number of INPs per mass of total organic carbon to that per mass of total organic matter, division by a conversion factor of 1.9 is applied. This value lies in the lower end of the range of factors recommended by Turpin and Lim (2001) for non-urban cites, and is chosen since we are only considering water insoluble organics, which are associated with lower carbon-to-molecule conversion in their study. Subsequent publications which investigated air borne sea spray aerosols

produced in laboratory settings (DeMott et al., 2016; McCluskey et al., 2017) have, however, indicated much lower ice nucleation efficiencies than that described by Wilson et al. (2015). Therefore a sensitivity study is also performed by producing a fit





to data published in DeMott et al. (2016). Both parametrisations are extrapolated to cover the entire temperature range relevant for mixed-phase clouds (-35 °C to 0 °C in ECHAM6-HAM2), and applied equally to accumulation and coarse mode MOA. For comparison, the parametrisations are plotted together with the $n_s$-based parametrisation of Niemand et al. (2012) for dust in Fig. 2. Orders of magnitude weaker ice-activity of MOA compared to dust can be noted, but MOA could still be important

5 in more remote regions where dust concentrations are low.

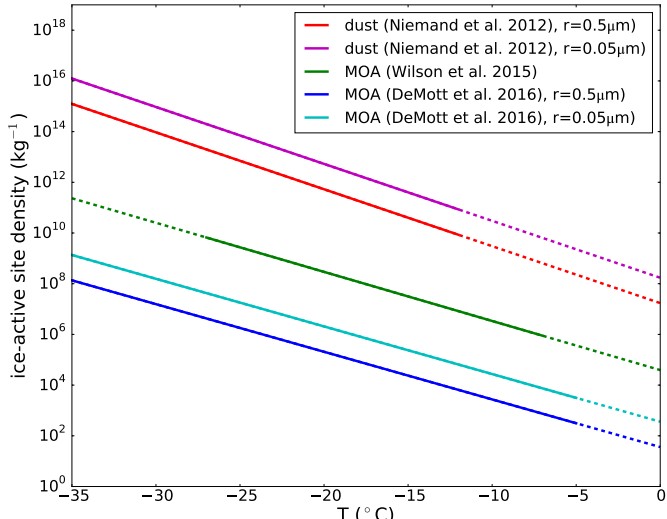

**Figure 2.** Ice-active site density per unit mass ($n_m$) of the Wilson et al. (2015) parametrisation and of the fit to the DeMott et al. (2016) data, for MOA, as well as the Niemand et al. (2012) parametrisation for dust. The Wilson et al. (2015) parametrisation is converted from INP number per total organic carbon mass to INP number per MOA mass by dividing by the conversion factor of 1.9. The DeMott et al. (2016) fit and Niemand et al. (2012) parametrisation are converted from the original representation of ice-active site density per unit surface area ($n_s$) by division by their respective density and multiplication by the spherical surface-to-volume ratio using the two extremes in accumulation mode median radius in our model. The inverse dependence of the ratio on the radius induces higher ice activity of the smaller particle when converting from $n_s$ to $n_m$. Solid lines represent the range in which the parametrisations are valid, and dotted lines represent temperature ranges where the parametrisations are linearly extrapolated.

The surface active site density ($n_s$) approach described in Connolly et al. (2009) is extended to consider active site density per mass ($n_m$), and applied to calculate a frozen fraction ($FF$) given the mean particle mass ($m_{MOA}$) and temperature. This is then multiplied by the number concentration of MOA immersed in cloud droplets ($N_{MOA,imm}$), such that the number of drops frozen per time step ($N_{frozen}$) is

$$N_{frozen} = N_{MOA,imm} \times FF$$

$$= N_{MOA,imm} \times [1 - \exp(-m_{MOA} \times n_{m,MOA})]. \tag{2}$$





$N_{\text{MOA,imm}}$ is defined as

$$N_{\text{MOA,imm}} = N_{\text{TOT,act}} \times \left( \frac{V_{\text{MOA}}}{V_{\text{TOT}}} \right)^{\frac{2}{3}} \tag{3}$$

following Hoose et al. (2008), where $V_{\text{MOA}}$ is the total volume of MOA in the mode calculated by dividing the mass by the density, and $V_{\text{TOT}}$ is the summed volume of all species in the internally mixed mode. $\left( \frac{V_{\text{MOA}}}{V_{\text{TOT}}} \right)^{2/3}$ is therefore a surface area

fraction which considers that although the species are internally mixed in the mode, not every particle will contain MOA. A surface area fraction is used as this is the relevant property for ice nucleation. $N_{\text{TOT,act}}$ is the number of aerosol particles in the internally mixed mode that can be activated to cloud droplets under current conditions, as calculated following Abdul-Razzak and Ghan (2000). This is equal to the actual CDNC only if the cloud cover or liquid water content in the grid box increased from the last time step, and only if the newly activated number is greater than the previous CDNC (Lohmann et al., 2007).

Further pertaining to Eq. (2), the mean mass of MOA in the size mode ($m_{\text{MOA}}$) is obtained by dividing the total mass of MOA in the mode by the total number scaled by the surface area fraction as defined above, and $n_{\text{m,MOA}}$ is the temperature-dependent number of active sites per mass, calculated using the Wilson et al. (2015) parametrisation. A slight modification is required for the fit to data from DeMott et al. (2016), which expresses the number of active sites per surface area instead of per mass. The mean surface area of MOA per particle is then defined as

$$s_{\text{mean}} = 4\pi \bar{r}^2 \exp \left( 2\ln^2 \sigma \right) \left( \frac{V_{\text{MOA}}}{V_{\text{TOT}}} \right)^{2/3}, \tag{4}$$

where $\bar{r}$ is the median radius of all particles in the mode and $\sigma$ is the standard deviation of the lognormal distribution, which is a size mode-dependent constant. The $s_{\text{mean}}$ is then multiplied by $n_{\text{s}}$ in the calculation for $FF$.

One problem with the above method of determining heterogeneous ice nucleation is that, in ECHAM-HAM, aerosols are not removed due to activation. Rather, in-cloud wet removal only occurs due to precipitation in the form of rain or snow. This

leads to possible repeat-freezing of the same aerosols across time steps. Indeed, the active site density approach of Connolly et al. (2009) represents the integrated number of ice crystals that can be frozen when the temperature drops from 0 °C to the current temperature, which would overestimate freezing if the full range of temperature drop from 0 °C is assumed at each time step. One method to address this is to subtract the ice crystal number concentration (ICNC) from the previous time step from the newly activated number, such that only when the latter is greater than the former, does ICNC change due to heterogeneous

freezing. This method has the drawback that it assumes that all ice crystals in the mixed-phase temperature range are produced through heterogeneous nucleation. In fact, the largest contributor to mixed-phase ICNC in our model has been found to be sedimentation from cirrus clouds, which can lead to suppression of contributions from heterogenous freezing (Ickes et al., in prep.). Thus in the case that the above method does not lead to an increase in ICNC, a second method is applied where $n_{\text{frozen}}$ calculated using the previous time step's temperature is subtracted from that calculated using the current temperature, such

that new ice crystals are produced if the temperature decreased since the last time step. This second method, in turn, does not consider transport of aerosols or changes in moisture between time steps and does not have memory beyond the previous time step. A combination of both methods is therefore applied to achieve a best estimate of the immersion freezing rate.





## 2.3 Simulations

Summarized in Table 2 is a range of sensitivity runs nudged to the same meteorology from March 2002 to May 2009 in order to investigate the impact of different set-ups on the MOA distribution while minimizing influences from internal variability. The nudging period is chosen to correspond to the maximum period covered by the MOA concentration measurements performed

at the two observational sites, to be described in Sect. 2.4. As mentioned previously, sensitivity to the chlorophyll concentration is investigated by replacing the SeaWiFS observations with the CMIP5 model mean outputs, while sensitivity to the sea salt emission is studied by using two different parametrisation schemes. Aside from the default set-up, the Guelle et al. (2001) parametrisation, which was the default sea salt emission set-up in a previous version of ECHAM-HAM and has a much higher emission rate, is also tested.

In most simulations, the Rinaldi et al. (2013) parametrisation for OMF is used, for it was found to fit best to observations when calculated offline using ECHAM6.3-HAM2.3 outputs as mentioned in Sect. 2.2.1 and shown in the appendix. This type of offline calculations, however, does not allow for proper consideration of particle size dependence, which is included in various size-resolved parametrisation schemes (e.g. Gantt et al., 2011; Long et al., 2011). Thus an additional sensitivity study is performed by using the size-resolved MOA emission parametrisation from Long et al. (2011), which also provides

a consistent emission scheme for both sea salt and MOA. It should be noted, however, that in the control set-up and most simulations following the same sea salt emission scheme, the total sea spray emitted according to the Long et al. (2011) parametrisation, which includes both sea salt and organic matter, is used to represent the sea salt emission. MOA mass is then emitted additionally and separately while the sea salt emission is kept untouched. To be consistent with the original intention of the parametrisation, the total sea spray is split between sea salt and MOA components in the size-resolved MOA simulation

where both species are emitted following Long et al. (2011). This would therefore also lead to changes in the sea salt emission rates.

Lastly, an additional simulation is performed where the MOA emission following Rinaldi et al. (2013) is doubled, one where the MOA emission is doubled and the ice-activity according to Wilson et al. (2015) is scaled up by 100, and another one where MOA is not emitted at all. The rationale behind the first simulation will be shown and discussed in Sect. 3.2, while that behind

the latter two will be discussed in Sect. 3.4.3..

Following the nudged runs, seven free-running sensitivity simulations are performed as listed in Table 3. The MOA emission set-up follows the "2xctlMOA" simulation, and is chosen based on the nudged run which best compares to observations, as will be discussed in Sect. 3.2. Each set-up is run for ten years (plus three months of spin up) with fixed year-to-year external forcing, and a ten-year mean is used during analysis to account for internal variability. As the goal of these free simulations is to

investigate the impact of ice nucleation by MOA and its climate feedback, focus is placed on the ice nucleation parametrisations.

MOA ice nucleation rates are studied by using the Wilson et al. (2015) parametrisation and a fit to data from DeMott et al. (2016), which has an ice-active surface site density that is around two to three orders of magnitude lower when converted to the same units. To test the potential impact of a highly ice-active MOA, another simulation is performed by increasing the ice activity of the Wilson et al. (2015) parametrisation by two orders of magnitude. Two different immersion freezing





**Table 2.** List of nudged simulations. In bold are fields which are changed from the control (ctl) set-up. Fields marked with dashes (-) are not relevant for the set-up.

| Name | MOA emission | MOA ice nucleation | Chlorophyll | Sea salt emission |
|---|---|---|---|---|
| ctl | Rinaldi et al. (2013) | Wilson et al. (2015) | SeaWiFS | Long et al. (2011) + Sofiev et al. (2011) |
| CMIP5chl | Rinaldi et al. (2013) | Wilson et al. (2015) | **CMIP5 mean** | Long et al. (2011) + Sofiev et al. (2011) |
| GuelleSS | Rinaldi et al. (2013) | Wilson et al. (2015) | SeaWiFS | **Guelle et al. (2001)** |
| LongMOA | **Long et al. (2011)** | Wilson et al. (2015) | SeaWiFS | Long et al. (2011) + Sofiev et al. (2011) |
| 2xctlMOA | **Rinaldi et al. (2013) $\times$ 2** | Wilson et al. (2015) | SeaWiFS | Long et al. (2011) + Sofiev et al. (2011) |
| MOA100ndg | **Rinaldi et al. (2013) $\times$ 2** | **Wilson et al. (2015) $\times$ 100** | SeaWiFS | Long et al. (2011) + Sofiev et al. (2011) |
| noMOAndg | - | - | - | Long et al. (2011) + Sofiev et al. (2011) |

parametrisations for dust, which is the only other heterogeneous freezing candidate in our model, are also tested. Control simulations are performed using a physically-based classical nucleation theory (CNT) parametrisation described in Ickes et al. (2017). Properties of montmorillonite as the ice nucleating dust type and an ice nucleation time integration of the first 10 seconds of each time step are chosen. Another set of simulations is done with the Niemand et al. (2012) ice-active surface site density ($n_s$)-based dust freezing parametrisation, which provides a more straight-forward comparison to the ice-active site density parametrisation of MOA. The $n_s$ parametrisation is extrapolated as is consistent with that of MOA.

Two simulations are done where MOA is emitted but not allowed to nucleate ice, each with a different dust freezing parametrisation (CNT vs. $n_s$), and finally one simulation is set up where MOA is not emitted in the model at all. Analysis of results from all of the above mentioned simulations are discussed in Sect. 3.

**Table 3.** List of 10-year free-running simulations. Fields marked with dashes (-) are not relevant for the set-up.

| Name | MOA emission | MOA ice nucleation | Dust ice nucleation |
|---|---|---|---|
| MOA | Rinaldi et al. (2013) $\times$ 2 | Wilson et al. (2015) | CNT (Ickes et al., 2017) |
| MOA100 | Rinaldi et al. (2013) $\times$ 2 | Wilson et al. (2015) $\times$ 100 | CNT (Ickes et al., 2017) |
| MOADeMott | Rinaldi et al. (2013) $\times$ 2 | based on DeMott et al. (2016) | CNT (Ickes et al., 2017) |
| noMOAfrz | Rinaldi et al. (2013) $\times$ 2 | - | CNT (Ickes et al., 2017) |
| MOA100ns | Rinaldi et al. (2013) $\times$ 2 | Wilson et al. (2015) $\times$ 100 | $n_s$ (Niemand et al., 2012) |
| noMOAfrzns | Rinaldi et al. (2013) $\times$ 2 | - | $n_s$ (Niemand et al., 2012) |
| noMOA | - | - | CNT (Ickes et al., 2017) |



## 2.4    Comparison to observations

Very limited long-term observations of MOA are available for validation of the model results. The two main sites with available data are Mace Head in Ireland and Amsterdam Island in the southern Indian Ocean. Measurement data from Mace Head spans the time period of 2002 to 2009 (Rinaldi et al., 2013), while that from Amsterdam Island covers the years of 2003 to 2007

(Sciare et al., 2009). For comparison with observations, model simulations nudged towards the meteorology of the respective measurement periods are used. Due to the limited spatial coverage of satellite-observed chlorophyll concentrations over single-month time spans, multi-year monthly mean chlorophyll measurements from SeaWiFS over the time period from March 2002 to May 2009 are used repeatedly for all simulation years. Should the dependence of MOA emissions on chlorophyll concentrations be strong in reality and the chlorophyll concentrations be highly variable from year to year, this may have contributed to biases

and inconsistencies in the modelled concentrations when compared to observations. Results from the comparisons are shown in Sect. 3.2.

## 3    Results

### 3.1    Distribution of MOA concentrations

A summary of MOA and sea salt annual emissions and global burdens from the various simulations is shown in Table 4.

A dependence on both the choice of chlorophyll concentration data and the sea salt emission scheme can be observed, as expected. Notably, a roughly doubling of the MOA burden resulted from the doubling of the Rinaldi et al. (2013) MOA flux, indicating a linear dependence of MOA burden on the emission rate. The same cannot be said, however, when the emission parametrisations are changed (for instance, when comparing the "ctl" simulation with "GuelleSS"), which resulted in changes in the spatial distribution of emitted MOA and thus diverging changes in emission and burden.

All emission rates are roughly in line with other quotes in the literature, which consider varying degrees of size resolution (e.g. Langmann et al., 2008; Spracklen et al., 2008; Long et al., 2011) and span a wide range of around 5 to 55 Tg y$^{-1}$ of organic matter when applying an organic carbon-to-organic matter conversion of 1.9. While studies that emit MOA in all size modes and consider both primary and secondary sources can obtain MOA fluxes of over 140 Tg y$^{-1}$ (Roelofs, 2008), most studies quote emission rates of less than 20 Tg y$^{-1}$, especially in the submicron size mode (e.g. Vignati et al., 2010; Lapina

et al., 2011; Gantt et al., 2011). On the global annual average, the mass emission of sub-micron primary MOA is less than 3 % of the total sea salt mass emission in all our simulation set-ups.

Annual mean global emission distributions and zonal mean cross sections of the mass concentration are shown in Fig. 3. Using the Rinaldi et al. (2013) parametrisation (all but "LongMOA"), the spatial pattern of MOA emission largely follows that of the chlorophyll concentration. For the simulations using SeaWiFS chlorophyll maps (all but "CMIP5chl"), MOA emissions

peak in coastal and equatorial upwelling regions. Due to higher surface wind speeds, a lower emission rate is found in mid-to-high latitude open ocean regions despite having chlorophyll concentrations comparable to the tropics. Notably and contrary to expectations based on previous literature (e.g. Burrows et al., 2013; Yun and Penner, 2013; Vergara-Temprado et al., 2017), we





**Table 4.** Annual global mean MOA and SS emissions and burdens from nudged simulations. The ratio of MOA to SS emission rates (MOA/SS emission) is a global area-weighted average of the ratio calculated at individual grid boxes. Note that MOA is only emitted in the accumulation mode while SS is emitted in both accumulation and coarse modes.

| Name | MOA emission (Tg y$^{-1}$) | MOA burden (Mg) | SS emission (Tg y$^{-1}$) | SS burden (Mg) | MOA/SS emission (%) |
|------|------|------|------|------|------|
| ctl | 9 | 67 | 1002 | 4129 | 1.4 |
| 2xctlMOA | 17 | 132 | 1002 | 4136 | 2.8 |
| CMIP5chl | 17 | 119 | 1002 | 4139 | 2.5 |
| GuelleSS | 10 | 63 | 5196 | 10558 | 0.26 |
| LongMOA | 10 | 63 | 956 | 3959 | 1.0 |

do not obtain a high concentration of MOA in the Southern Oceans with the Rinaldi et al. (2013) emission. When using CMIP5 mean chlorophyll concentrations, emissions from coastal upwelling regions are reduced while those from the equatorial upwelling become more pronounced. Much higher MOA emission rates can also be found in mid-to-high latitude open waters and the Southern Ocean (mainly occurring during the respective hemispheric summer months), as was observed in the chlorophyll

concentrations (Fig. 1). An anomalously high emission rate of MOA off the coast of the Arabian Peninsula in Boreal summer, observable even in the annual mean in all simulation set-ups, is mostly due to the Southwest Monsoon-associated ocean upwelling and particularly strong sea salt fluxes. The annual mean signal is weaker in the "LongMOA" simulation, which exhibits a weaker dependence on the wind speed and chlorophyll concentration. This leads to peak MOA emissions off the coast of the Arabian Peninsula only during Boreal summer, while relatively high emission rates are also present during Boreal autumn

and winter in the other simulations, corresponding to the chlorophyll map. The secondary peak in ocean productivity during winter months is associated with deep water mixing caused by colder air blowing over the water surface during the Northeast Monsoon season (Mann and Lazier, 2005; Wiggert et al., 2000).

     The annual mean MOA burden mostly mirrors the emission pattern, with notable accumulations over source regions that are subject to limited precipitation washout. The peak burden in the equatorial South Pacific found in all simulations, for instance,

can be associated with the dry zone related to the South Pacific convergence zone that is largely caused by orographically induced subsidence (Takahashi and Battisti, 2007), as well as contributions from subsiding branches of the Hadley and Walker circulations. On the other hand, high emission rates in the northern North Pacific and North Atlantic Oceans as well as along the Southern Ocean in the "LongMOA" simulation are not reflected in the annual mean burden, due to washout along the storm tracks.

In the zonal mean cross-section, MOA mass is mainly concentrated in the lower altitudes below 700 hPa, and is in general not transported very high up into the atmosphere, as can be expected since MOA is mainly emitted from relatively calm waters. Despite this, non-negligible amounts of MOA still reach mixed-phase temperatures, especially in sub-polar regions.





**Figure 3.** Multi-annual mean emissions (first column), burdens (second column), and zonal mean cross-sections (third column) of MOA from the various nudged simulations described in Table 2 for the period from March 2002 to May 2009. Contour lines in the zonal mean plots are zonal mean isotherms in °C in the mixed-phase temperature range. Red stars in the first plot on the upper left indicate locations of Mace Head in the Northern Hemisphere and Amsterdam Island in the Southern Hemisphere, where long-term observations of MOA are available.



All simulation set-ups produced similar patterns, with some having a more poleward extend of higher MOA concentrations than others, depending on the emission rates.

## 3.2 Comparison of MOA concentrations to the observed annual cycle

A comparison of the monthly mean MOA concentrations simulated using the various nudged set-ups listed in Table 2 to the observations at Amsterdam Island and Mace Head is shown in Fig. 4. Notably, the offline-calculated MOA concentration using the same set-up as the control simulation, which was used for choosing the OMF parametrisation as described in Sect. 2.2.1, is also plotted. It can be observed that even with the same parametrisation set-up, offline calculations yielded a stronger seasonal cycle than online calculations. Possible reasons for this deviation include errors in estimating the source regions (since the back trajectories are not explicitly computed using our model), seasonal variations in the aerosol source regions that is not considered in the offline calculations, and a lack of consideration for depletion and sedimentation during transport of MOA to the measurement site. As most MOA emission parametrisations are developed using similar offline methods, it may be worthwhile to note the possible deviation for future parametrisations. Due to the rather low bias of the control simulation, an additional simulation is set up where the control MOA emission using the Rinaldi et al. (2013) OMF parametrisation is increased two times (green curve in Fig. 4), and a better agreement to observations is obtained, despite a rather high bias at Mace Head in March and a low bias in January at Amsterdam Island. This increased emission is thus used as the standard set-up for all subsequent free-running 10-year simulations.

In examining other simulations with various sea salt or MOA emission set-ups, we found a general and consistent underestimation of MOA concentrations at both stations. One exception is the simulation using the Guelle et al. (2001) sea salt parametrisation, which produced reasonable MOA concentrations in Mace Head and hit the lower range of observations at Amsterdam Island except in the first four months of the year. The control simulation underestimated MOA concentrations at both stations. Simulation with MOA emitted following the size-resolved Long et al. (2011) parametrisation, on the other hand, was not able to reproduce the annual cycle of MOA concentrations well, despite yielding higher concentrations in Amsterdam Island. By replacing the SeaWiFS chlorophyll observations with CMIP5 mean modelled outputs, a significant underestimation of MOA concentrations resulted in Mace Head, while in Amsterdam Island, a relatively good fit to the observed annual cycle of MOA concentration is produced except for the months of October to December, where the CMIP5 models produced a widespread peak in chlorophyll concentrations in the Southern Ocean not observed by the satellite. This led to the decision to not use the CMIP5 mean chlorophyll map for the longer-term simulations, and points to the need for more MOA measurement sites and improvements in the simulation of chlorophyll concentrations in ocean models, as well as a need for caution in future MOA-related model studies where modelled chlorophyll concentrations need to be used. Notably, however, the "CMIP5chl" simulation is the only simulation which is able to reproduce the strong peak in MOA concentrations during the austral summer months (DJF) in Amsterdam Island, which may point to an underestimation of the multi-year mean SeaWiFS chlorophyll concentrations in these months or to other missing marine organic sources not directly associated with chlorophyll.





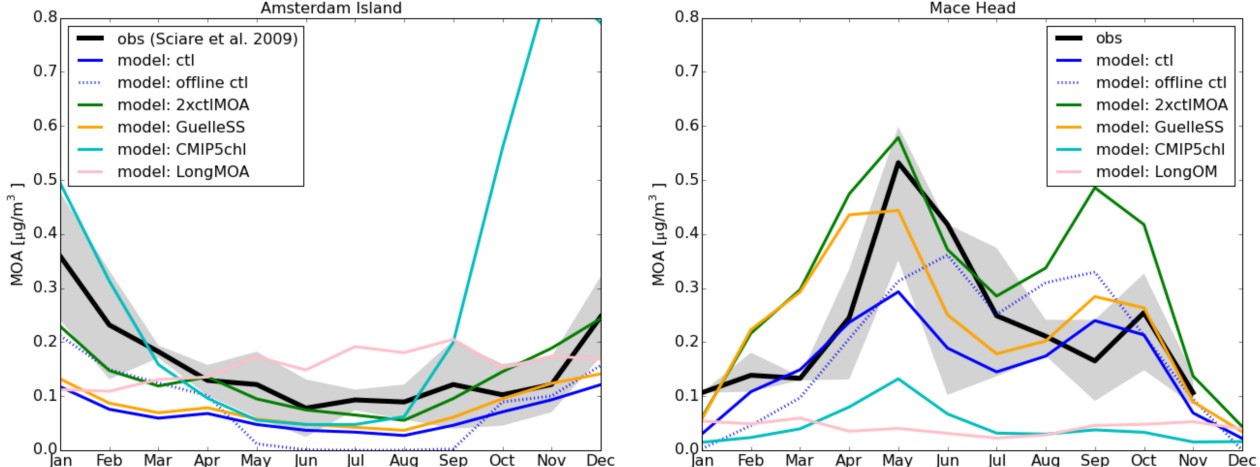

**Figure 4.** Monthly mean observed MOA concentrations at Amsterdam Island and Mace Head compared to model simulated outputs as described in Table 2. The dotted blue curve ("offline ctl") corresponds to an offline calculation of MOA concentration based on the same parametrisation as the control case. Shaded area corresponds to plus and minus one standard deviation of the observed mean for the Amsterdam Island case, while for Mace Head, it corresponds to the 25th to 75th percentile. In both cases, the bold black line is the monthly observational mean. Note that no measurements are available for Mace Head in December. Model outputs are averaged over the entire nudging period (March 2002 to May 2009) for comparisons to observations at Mace Head, and averaged over the period of May 2003 to November 2007 for Amsterdam Island, as according to the measurement campaigns.

### 3.3 Heterogeneous ice nucleation

Ice nucleation rates due to immersion freezing of MOA and dust and their respective frequencies of occurrence when applying the various parametrisations are shown in Fig. 5. One clear observation is the at least five orders of magnitude difference between the peak freezing rates of dust and MOA. Aside from this, freezing occurs with the same frequency for all active
5 site density schemes (contour lines in Fig. 5). Freezing calculated using this parametrisation approach occurs whenever the environmental condition is conductive for mixed-phase freezing (the presence of supercooled liquid droplets with ambient temperature between 0 and -35 °C), the relevant ice-active aerosol species is present in sufficient amounts (relative to the respective ice activity), and the considerations for re-freezing as described in Sect. 2.2.2 is fulfilled. This indicates that in most cases, MOA and dust are both present in sufficient amounts and thus freezing occurs for both species if the environmental
10 factors allow.

Such direct conclusions cannot be drawn from comparison with the CNT results, however. While the CNT-based dust freezing scheme produces lower freezing occurrence frequency overall and especially in the warmer temperatures compared to MOA (Fig. 5), a similar difference in freezing occurrence frequency can also be noted between results from the two dust





freezing schemes, which point to the parametrisation as the main reason behind the difference. The sharp decrease in freezing occurrence at warmer temperatures following Ickes et al. (2017) when compared to the results using the Niemand et al. (2012) dust parametrisation indicates a faster decrease of the dust ice activity with increasing temperature for the former set-up, which is shown in Fig. 6. Indeed, the $FF$ of 0.5 $\mu$m radius particles following Ickes et al. (2017) quickly drops below that

following Niemand et al. (2012) at around -31 °C, and below that of MOA following Wilson et al. (2015) at around -25 °C. By around -20 °C, even with the maximum monthly mean immersed dust concentration of the order of 100 cm$^{-3}$, only around ten droplets will freeze in one cubic kilometre of air. Without sufficiently large dust particles and/or sufficiently high number concentrations, the Ickes et al. (2017) CNT parametrisation will thus not lead to much ice nucleation occurrence in the warmer mixed-phase temperatures. Interestingly, a surface active site density approach for montmorillonite (consistent in dust type with

the CNT parametrisation), which is also compared in Fig. 6, can be noted to have a less steep slope than the CNT approach, though still with a faster decrease in $FF$ than that from Niemand et al. (2012). As the Niemand et al. (2012) parametrisation considers a mixture of dust mineral types, this indicates that the difference in freezing rates between the Ickes et al. (2017) CNT parametrisation and Niemand et al. (2012) may be the consequence of both a difference in parametrisation method (CNT vs. $n_s$) and the considered dust type (montmorillonite vs. Saharan dust). A CNT-based parametrisation for MOA cannot be

formulated, however, due to the lack of measurement data. It is therefore impossible to conclude how much of the difference in the frequency and regions of occurrence between freezing by MOA and CNT-parametrised dust is related to the different INP species and how much is simply due to differences in the parametrisation approach.

**Immersion freezing by dust**

Dust freezing rates following CNT (Ickes et al., 2017) and the surface active site density approach using the Niemand et al.

(2012) parametrisation show consistent results in the spatial distribution and magnitude of peak values mainly in the colder mixed-phase temperature range. The Niemand et al. (2012) parametrisation, which is extrapolated for temperatures warmer than -12 °C, leads to more frequent freezing occurrence, especially notable at warmer temperatures, albeit with significantly lower freezing rates when compared to that in colder regions. This is associated with the differing slopes of the two parametrisations, as discussed in the previous section and shown in Fig. 6. When expressed as the temperature at which a $FF$ of 0.001

is reached, this translates to -21 °C for Niemand et al. (2012) and -28 °C for Ickes et al. (2017), assuming spherical aerosols of radius 0.5 $\mu$m.

**Immersion freezing by MOA**

The MOA freezing rate scales proportionally to the ice-active site density, with freezing rates increasing by around two orders of magnitude (depending on the aerosol size) when using the Wilson et al. (2015) parametrisation compared to the DeMott

et al. (2016) parametrisation, and when scaling up the ice activity of the Wilson et al. (2015) parametrisation by 100 compared to the standard version. The freezing onset temperatures of MOA for the same conditions as described above for dust are, respectively, -59 °C, -40 °C, and -29 °C for the parametrisation in the "MOADeMott", "MOA", and "MOA100" set-ups.



**Figure 5.** Seasonal and zonal mean freezing rates due to MOA and dust during freezing occurrence. Contour lines denote the frequency with which freezing occurs. All plots are averages from the 10-year free simulations. Only rates where the freezing occurs more often than 0.1 % of the time are plotted.





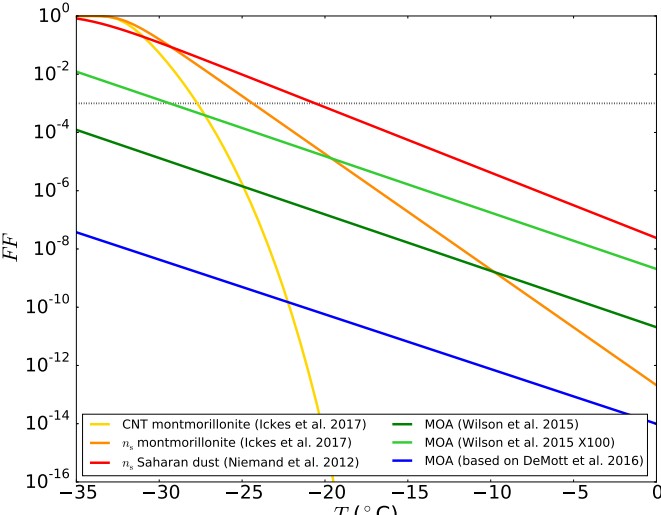

**Figure 6.** Frozen fraction ($FF$) versus temperature ($T$) of the various parametrisations used for dust (montmorillonite and Saharan dust) and MOA in this study. Spherical aerosols of radius 0.5 $\mu$m and an ice nucleation time of 10 seconds for the CNT method are assumed. In addition to the parametrisations previously discussed in the text, an additional $n_s$-based parametrisation for montmorillonite is also plotted following Ickes et al. (2017). The dotted black line indicates the 0.001 $FF$ level.

### 3.3.1 MOA vs. dust as INP

The immersion freezing rate, as described by Eq. 2, is calculated by multiplication of the number of aerosol particles available for freezing, which depends on the abundance and distribution of the aerosols as defined in Eq. 3, and the $FF$, which depends on the property of the aerosols (size- and temperature-dependent ice activity). A decomposition of these two components for
MOA and dust is shown in Fig. 7. Here it can be noted that the number concentration of immersed MOA and dust span similar orders of magnitude in the accumulation mode, while in the coarse mode the abundance of dust can be up to two orders of magnitude greater. The $FF$, on the other hand, shows a more significant four orders of magnitude difference between dust and MOA regardless of the size mode. Thus the temperature-dependent aerosol ice activity can be attributed as the main controlling factor behind the number concentration of nucleated ice crystals as compared to the availability of the particles. This can also be
concluded by noting the small change in MOA freezing rates when different sea salt emission parametrisations or chlorophyll maps are used (not shown). Nonetheless, the larger amount of MOA near the surface can contribute to higher ice nucleation rates in polar near-surface regions despite the warmer temperature.

   So far, only seasonal or annual mean freezing rates have been shown. However, monthly mean dust concentrations in the air could be dominated by episodic dust events which would mask potential contributions from MOA during periods of low
dust concentrations. Thus online diagnostics of the time frequency when the freezing rate of MOA is greater than that of dust is performed for cloudy mixed-phased grids containing supercooled droplets and shown in Fig. 8. Different combinations of MOA and dust freezing parametrisations are investigated.





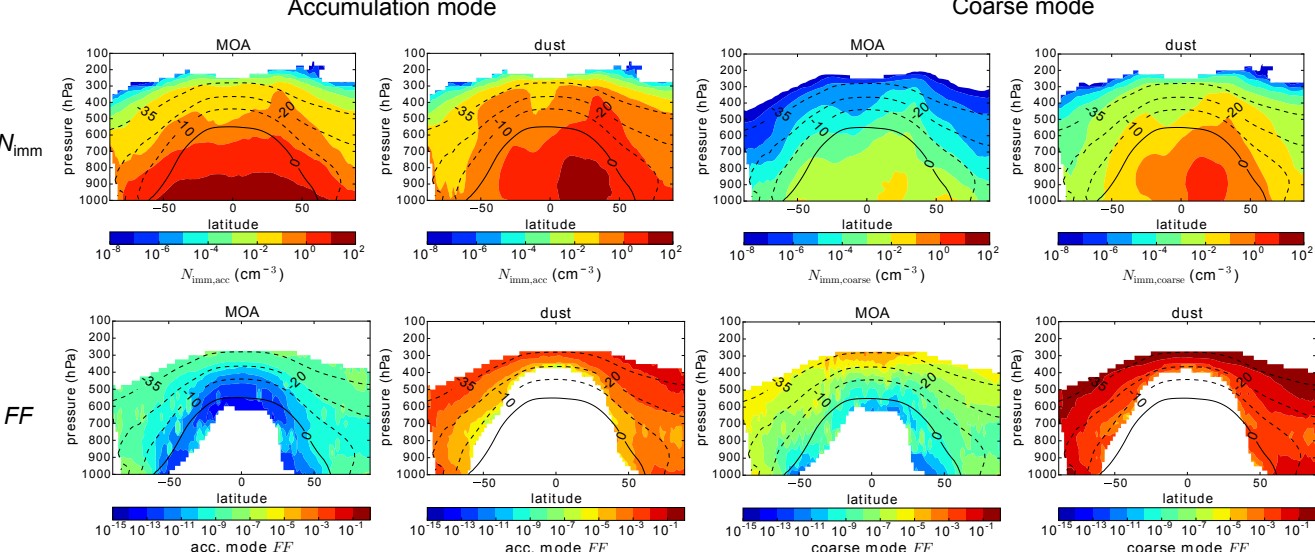

**Figure 7.** Multi-annual, zonal mean number of accumulation [leftmost two columns] and coarse [rightmost two columns] mode particles immersed in droplets ($N_{\mathrm{imm}}$; first row) and the frozen fraction ($FF$; second row) for MOA [first and third columns] and dust [second and fourth columns], from the free "MOA" simulation with the Wilson et al. (2015) freezing parametrisation for MOA and CNT for dust. Contour lines denote isotherms in the mixed-phase temperature range. $FF$ is only plotted where the freezing occurs more frequently than 0.1 % of the time.

When comparing between the diagnostic results with varying set-ups of freezing by MOA, no systematic differences are present. In particular, only a very slight increase in the frequency of occurrence resulted from a two orders of magnitude increase in the MOA ice-activity, as can be noted from comparison between the "MOA" and the "MOA100" simulations. When comparing between the "MOA" and the "MOADeMott" simulations, a more noticeable decrease in frequency of occurrence can

be noted due to the lower MOA ice activity of the DeMott et al. (2016) data. This can be attributed to the differences/similarities between the frequency of MOA freezing occurrence of the "MOA Wilson" and "MOA DeMott"/"MOA WilsonX100" simulations shown in Fig. 5 (contours).

While the choice of MOA freezing parametrisation does not have a significant qualitative influence on the result of such diagnostics, that of the dust ice nucleation parametrisation plays a significant role. This is due to the much lower freezing

frequency from the CNT-based approach, especially in warmer temperature regimes. When the $100 \times$ Wilson et al. (2015) parametrisation for MOA is combined with the Ickes et al. (2017) CNT parametrisation for dust, the contribution from MOA frequently dominated over that from dust over much of the warmer mixed-phase regions. When the Niemand et al. (2012) $n_{\mathrm{s}}$ parametrisation is applied for dust, however, MOA only becomes more important than dust in the warmest mixed-phase regions near the surface in polar regions and in the Southern Hemisphere low altitudes. This difference can be attributed to the

different slopes and dust freezing onset between the CNT and $n_{\mathrm{s}}$ parametrisations, as noted previously in Sect. 3.3.





Regardless of differences between different freezing parametrisation set-ups, MOA has been found to contribute to more freezing than dust during up to 20 to 70 % of the time in much of the mixed-phase cloud regions when using the Ickes et al. (2017) dust scheme, and up to 60 % near the surface in the Southern Hemisphere when using the Niemand et al. (2012) dust scheme. This is largely comparable to the values found by Vergara-Temprado et al. (2017), who examined the percentage

of days when the INP concentration at ambient temperatures from MOA is greater than that from K-feldspar. As their study also uses a $n_s$-based freezing parametrisation for dust, the most straightforward comparison would be with our "MOA100ns" simulation, in which case slightly lower freezing contributions from MOA are found in our results. This is especially notable in the Northern Hemisphere and in the higher altitudes in the Southern Hemisphere. Possible reasons for this include their choice of only considering freezing by a fraction of the dust (K-feldspar) instead of all dust types in our case, which decreases

the availability of dust particles in their study and leads to a more ready scavenging of the dust INP from the atmosphere due to the larger size of feldspar, as noted also by Vergara-Temprado et al. (2017). Additionally, the freezing parametrisations are not extrapolated to all mixed-phase temperatures in their study, and lastly, considerable differences in emission, partitioning, removal, and transport of the aerosols can also exist between models.

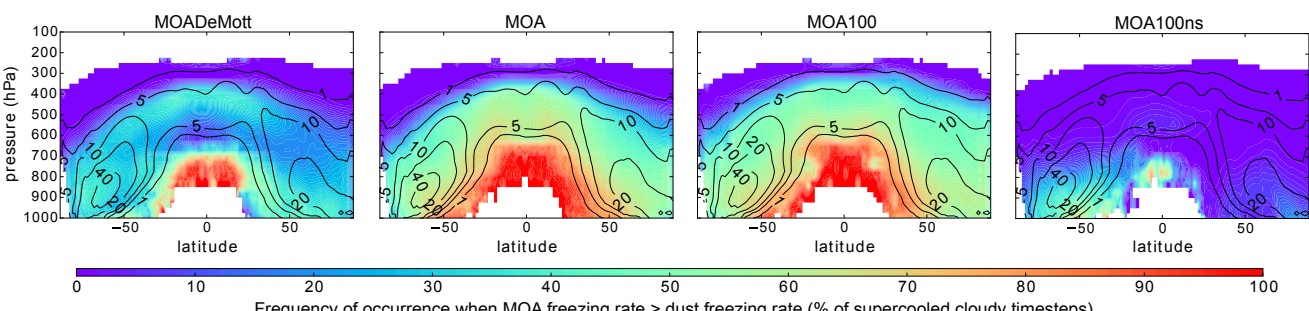

**Figure 8.** The annually and zonally averaged frequency of occurrence when the freezing contribution from MOA is greater than that from dust, diagnosed only for cloudy grid boxes containing supercooled droplets. Contour lines denote the frequency of occurrence (in %) of the above-mentioned favourable cloudy condition containing liquid droplets in the mixed-phase temperature range. All plots are from the respectively titled 10-year free-running simulations.

## 3.4 Impact on clouds and climate

### 3.4.1 Impact on clouds

INPs can impact clouds through freezing of supercooled liquid droplets and subsequent ice crystal growth at the expense of the remaining liquid drops, leading to glaciation of the cloud. The most direct impact of MOA as an INP would thus be expected in the cloud ice and liquid properties. This is shown in Fig. 9 as the annual mean in-cloud relative difference between one simulation where the expected impact of MOA acting as INP is greatest ("MOA100") and the corresponding simulation where

it is not allowed to initiate freezing ("noMOAfrz"). The most statistically significant change in cloud properties is found near




the surface over the Southern Ocean, with a decrease in the in-cloud zonal and annual mean ice crystal radius ($r_{\mathrm{eff,i}}$) by up to 3 to 9 %. This is associated in general with a non statistically significant increase in the ice cloud occurrence frequency as well as decreases in the in-cloud ice water content (IWC) and ice crystal number concentration (ICNC).

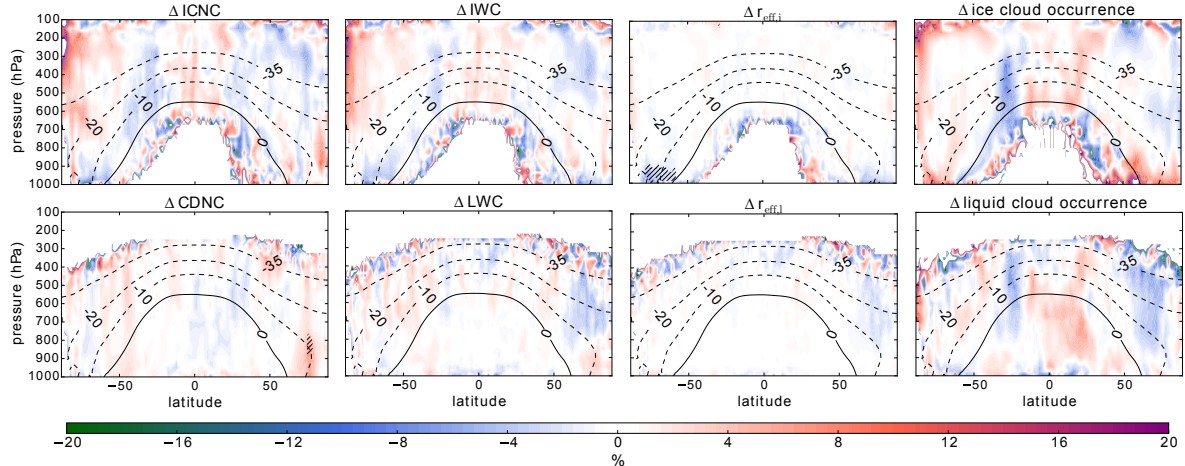

**Figure 9.** Annual and zonal mean relative change ("MOA100" minus "noMOAfrz" divided by the mean of the two) in ice crystal number concentration (ICNC), ice water content (IWC), ice crystal effective radius ($r_{\mathrm{eff,i}}$), ice cloud occurrence frequency, cloud droplet number concentration (CDNC), liquid water content (LWC), cloud droplet effective radius ($r_{\mathrm{eff,l}}$), and liquid cloud occurrence frequency. All values are in-cloud changes (i.e. during liquid/ice cloud occurrence, respectively). Contour lines are zonal mean temperatures in the mixed-phase range in °C. Hatched areas indicate statistical significance at the 95 % level following the Wilks (2016) method for controlling the false discovery rate for data with moderate to strong spatial correlations.

Changes in monthly mean cloud properties can be due to formation of new clouds and/or as a result of changes in pre-existing ones. To investigate the cause of the decrease in $r_{\mathrm{eff,i}}$, further diagnostics is performed by separating grid boxes where the monthly mean ice cloud occurrence increased (when comparing outputs from "MOA100" to those from "noMOAfrz") and where they decreased or stayed the same. The zonal and multi-annual mean $\Delta r_{\mathrm{eff,i}}$ is then diagnosed for these two cases separately and shown in Fig. 10. An increase in the time-mean cloud occurrence frequency is indicative of an increase in the cloud lifetime and/or an increase in the cloud initiation frequency. In the absence of stratiform precipitation (e.g. in the tropics at high altitudes), grid boxes with higher humidity can sustain larger ice crystals and for longer lifetimes due to slower depletion through sublimation. Once the ice crystals approach precipitation sizes, however, larger $r_{\mathrm{eff,i}}$ would be associated with shorter cloud lifetimes due to depletion through snowfall. Increases in the cloud initiation frequency, on the other hand, would decrease the time-averaged in-cloud $r_{\mathrm{eff,i}}$ as newly formed ice crystals tend to be smaller than those which have had a chance to grow through aggregation, etc.

In general, the $\Delta r_{\mathrm{eff,i}}$ in the two cloud occurrence cases cancels out, simply due to zonal variability and separation of grid boxes with different cloud occurrence changes. Changes that do not cancel out between the two cases can then be attributed to




the influence of MOA acting as an INP (Fig. 10a). This is the case over the Southern Ocean, where a decrease in $r_{\text{eff,i}}$ during increased ice cloud occurrence (Fig. 10b) is not reflected with a respective $r_{\text{eff,i}}$ increase in cases with decreased or constant ice cloud occurrence (Fig. 10c). It can be speculated, therefore, that MOA leads to a decrease in $r_{\text{eff,i}}$ over the Southern Ocean through an increase in ice crystal initiation (which would lead to increases in ice cloud occurrence frequency). Freezing of

cloud droplets and further depletion due to the Wegener-Bergeron-Findeisen process can then lead to reductions in the lifetime of liquid cloud droplets. In support of this hypothesis, a reduction in liquid cloud occurrence frequency in the same regions near the surface over the Southern Ocean, though not statistically significant, can be noted in Fig. 9. Since MOA can nucleate ice under warmer conditions, the newly formed crystals can also be accreted more easily by falling snow, which has a collection efficiency that increases with temperature. This would dampen the correlation between increases in the ice cloud occurrence

and decreases in the crystal radius, and can help explain the slight difference in the patterns of the two respective changes in Fig. 9.

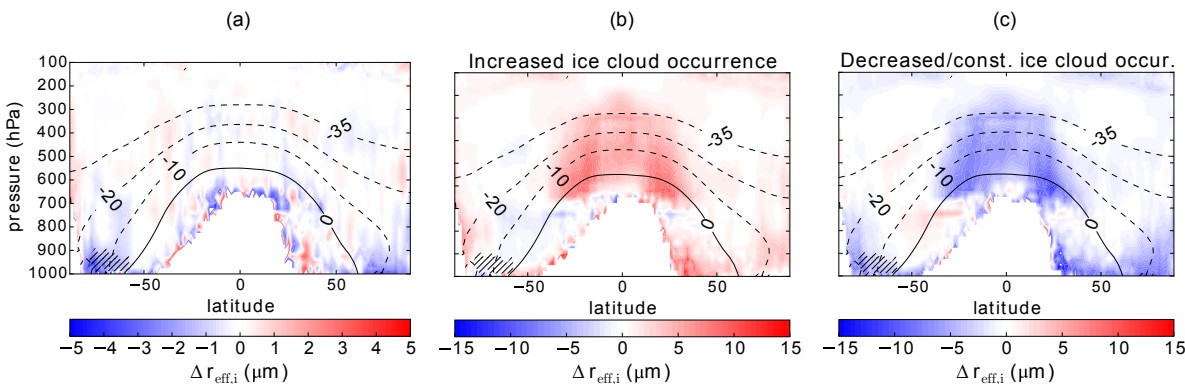

**Figure 10.** The left plot is equivalent to the $\Delta r_{\text{eff,i}}$ plot in Fig. 9, except in absolute terms. The rightmost two plots are diagnostics of the same property but separated into cases where the grid (not zonal) mean change in monthly mean ice cloud occurrence frequency is positive (b) and where the change is negative or neutral (c). Grids boxes which do not satisfy the case criterion are omitted from the respective zonal mean analysis. Hatched areas indicate where statistically significant changes are found without case separation (i.e. where significance is found in Fig. 9). Contour lines are zonal mean temperatures in the mixed-phase range in °C.

A mirrored decrease in $r_{\text{eff,i}}$ in the Arctic can also be noted, though without statistical significance and with less consistent corresponding changes in the ICNC and IWC (Fig. 9). The ice cloud occurrence frequency, on the other hand, increases strongly in these regions and may be better associated with the decrease in $r_{\text{eff,i}}$. The lack of a clear statistically significant

signal in the Northern Hemisphere can be attributed to the stronger zonal variability due to land-sea contrasts that is absent over the Southern Ocean. Moreover, the Wilks (2016) method for controlling the false discovery rate tends to underestimate the significance (Wilks, 2016), which may have also contributed to the lack of statistical significance in the Arctic.





In the cloud liquid properties, the only statistically significant change is a slight increase in the in-cloud cloud droplet number concentration (CDNC) over the Arctic (Fig. 9). This is, however, more likely dynamically induced, a topic which is discussed further in Sect. 3.4.3.

### 3.4.2 Impact on the TOA radiative balance

The change in the zonal mean top-of-atmosphere (TOA) radiative balance due to the emission and/or ice activity of MOA in the various free-running set-ups are shown in Fig. 11. When comparing our strongest MOA potential set-up ("MOA100") to one where MOA is not emitted at all ("noMOA"), the TOA net solar radiation decreases by 0.34 W m$^{-2}$ on the global area-weighted mean with the added MOA, while the net outgoing terrestrial radiation decreases by 0.07 W m$^{-2}$, leading to a net of 0.27 W m$^{-2}$ more total outgoing radiation at the TOA. The strongest signal of change can be observed over the Southern

Ocean and Antarctica, with a decrease in the outgoing radiation of up to 1.5 W m$^{-2}$ in the zonal mean. When decomposing the contribution into that from the emission of MOA ("noMOAfrz"-"noMOA") and that from MOA acting as INP ("MOA100"-"noMOAfrz"), neither process can be ruled out as a contributor to the change. This can include cooling at the surface due to the scattering direct effect of the emitted MOA and that due to the increased aerosol indirect effect on cloud radiative properties induced by MOA acting as INP, as well as further feedbacks through dynamics triggered by the two processes. In particular, the

signal from MOA emission increases toward the South Pole while that from MOA acting as INP is more consistent throughout the latitude bands over the Southern Ocean. Neither of these changes, however, are statistically significant by themselves (except for latitude bands right over Antarctica, for MOA emission), pointing to a synergistic effect of the two processes.

When investigating other free-running set-ups, no consistent result can be observed. No consistent pattern can be observed in the changes due to MOA acting as INP when the dust ice nucleation parametrisation is changed from Ickes et al. (2017)

to Niemand et al. (2012) (i.e. "MOA100"-"noMOAfrz" vs. "MOA100ns"-"noMOAnsfrz"). When examining the total changes without the two orders of magnitude increase in the Wilson et al. (2015) MOA ice-activity ("MOA"-"noMOA"), a weaker signal than due to MOA emission alone is observed in the TOA longwave (LW) radiation over the Southern Hemisphere. A similar pattern is present when compared to the stronger "MOA100"-"noMOA" case, but this may be due to both analysis having the same base simulation without MOA ("noMOA") which dominates the signal. The similarity becomes even more

elusive for the "MOADeMott"-"noMOA" analysis, especially for latitudes north of 50° S.

In the shortwave (SW), a statistically significant decrease in the net TOA incoming radiation of 0.3 to 1 W m$^{-2}$ over Antarctica and tropical regions south of the equator can be observed in association with the added MOA. However, the statistical significance disappears with any consideration of the freezing property except for the case with the weakest MOA ice-activity ("MOADeMott"), indicating more complicated feedback processes. This is further examined in the next section.

### 3.4.3 Impact on dynamics

The changes in the zonal mean aerosol and cloud properties together with the changes in TOA radiative balance due to the emission and ice activity of MOA in the "MOA100" set-up is shown in the left panel in Fig. 12. In particular, the SW aerosol forcing mirrors rather well the increase in aerosol optical depth (AOD, global mean increase of 0.006) at the various latitudes,



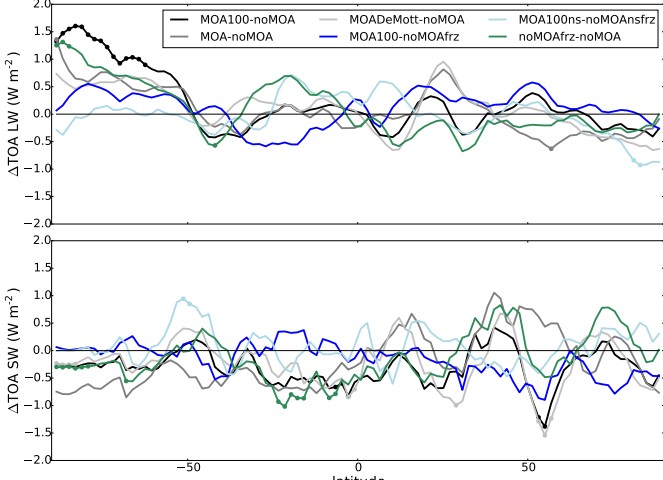

**Figure 11.** Zonal and multi-annual mean change in the top-of-atmosphere (TOA) solar (SW) and terrestrial (LW) radiative balance for the various free-running 10-year simulations. The black and gray curves indicate changes due to both the emission and ice nucleation of MOA, the blue-coloured curves correspond to changes stemming from MOA ice nucleation, and the green curve indicates changes due to the emission of MOA. It should be noted that the outgoing terrestrial radiation is defined to be negative, so a positive change is indicative of less outgoing radiation.

indicating an expected increase in scattering effect due to the added MOA. In the global mean, the TOA all-sky instantaneous aerosol forcing leads to $0.086$ W m$^{-2}$ less incoming radiation in the SW and $0.014$ W m$^{-2}$ less outgoing radiation in the LW, yielding a net cooling forcing of $0.072$ W m$^{-2}$ at the TOA by the addition of MOA and other feedback processes involving MOA acting as an INP that may affect the aerosol forcing.

The aerosol forcing does not, however, translate to changes in the TOA radiative balance, which is dominated by a decrease in outgoing radiation in the Southern Hemisphere high latitudes, as discussed in Sect. 3.4.2. Rather, the latter signal stems from feedback processes that led to a stronger zonal wind over the Southern Ocean and cooling over Antarctica. The specific pathway leading from the added MOA and INP to this feedback process, however, is not clear.

To further investigate possible causes for the cooling over Antarctica, the relevant simulations ("MOA100" and "noMOA")

are extended for an additional 10 years and the resulting changes shown in the middle panel in Fig. 12. Notably, the significant decrease in outgoing LW radiation found in the shorter simulation is much diminished. Instead, a more statistically significant increase in the TOA aerosol LW forcing (warming) can be observed in the Arctic. The slight overall increase in snowfall rate observed in the 10-year simulation is also further weakened. This is replaced with a dipole signal of a weak (on the orders of $0.001$ mm h$^{-1}$ or less) but statistically significant decrease in snowfall rate at $60°$ to $70°$ N and an approximately mirrored

increase in the Southern Hemisphere. Overall, however, we conclude that MOA emission and MOA acting as INP do not have significant impacts on the global radiative balance and climate variables, and signals observed in the 10-year simulations are likely to originate from internal variability. The only consistent changes between the 10-year and 20-year simulations are the



statistically significant increase in AOD and subsequent increase in negative aerosol SW forcing around 35° to 60° S, as well as greater general increases in AOD in the tropical latitudes.

Lastly, to suppress feedback processes through dynamics, analysis of nudged simulations with otherwise the same set-ups ("MOA100ndg" and "noMOAndg") are performed where the vorticity and divergence of the flow fields are nudged toward the
same meteorology for the two simulations. The changes in the climate and cloud properties are shown in the right panel in Fig. 12, where it can be noted that any changes in the examined properties are diminished and no significant impact of MOA on the model climate can be observed. The pattern of changes in the AOD and aerosol forcing are, however, consistent with other simulations.

## 4  Conclusions

In this study, a range of simulations are set up to investigate the emission and distribution of MOA on the global scale. Three different aspects that control the emission rate are tested, namely the sea salt emission parametrisation, the MOA emission parametrisation, and the chlorophyll map. A weaker dependence on the sea salt emission parametrisation is found compared to the choice of chlorophyll data and MOA emission parametrisation. In particular, the use of the CMIP5 mean modelled chlorophyll data to replace SeaWiFS observations leads to significant changes in the MOA spatial distributions. A cause for
this is the systematic overestimation of total chlorophyll concentrations in the Southern Ocean that is common among global ocean models (Le Quéré et al., 2016), and should be taken into account for future simulations using modelled chlorophyll concentrations. The vertical distribution of MOA, however, is relatively similar between simulations, with the mass mostly concentrated in the lower levels.

Following previous studies proposing MOA as a potentially important INP (e.g. Wilson et al., 2015; Vergara-Temprado
et al., 2017), contributions of MOA to heterogeneous ice nucleation is investigated. When compared to dust, MOA is found to nucleate five to ten orders of magnitude less ice crystals, depending on the choice of parametrisation, due to its weaker ice-activity. When compared to the CNT-based dust parametrisation for montmorillonite (Ickes et al., 2017), however, MOA commonly contributes to more freezing of liquid droplets during half of the favourable supercooled cloudy timesteps in the mixed-phase temperature range. A two orders of magnitude decrease in the MOA ice activity, i.e. without scaling up the Wilson
et al. (2015) parametrisation by 100, does not significantly affect the results. On the contrary, the use of the fit to data from DeMott et al. (2016), which indicates a further two orders of magnitude weaker MOA ice activity, brings the frequency of times when MOA nucleates more ice than dust closer to 30 % or less. This indicates a possible threshold where the droplet freezing rate due to background dust during non-dust events may lie somewhere between the "MOADeMott" and the "MOA" set-ups. On the other hand, when applied together with the Niemand et al. (2012) parametrisation for dust, MOA only contributes to
more heterogeneous ice nucleation than dust during up to 10 % of the time near the surface in the Arctic and around 30 to 40 % of the time in the Southern Hemisphere low altitudes, where the mass concentration of MOA is higher and where the dust concentration is lower due to the hemispheric dependence of dust emissions that favours the Northern Hemisphere. The difference between the comparisons to the two different dust parametrisations mainly stems from their differing rates of $FF$





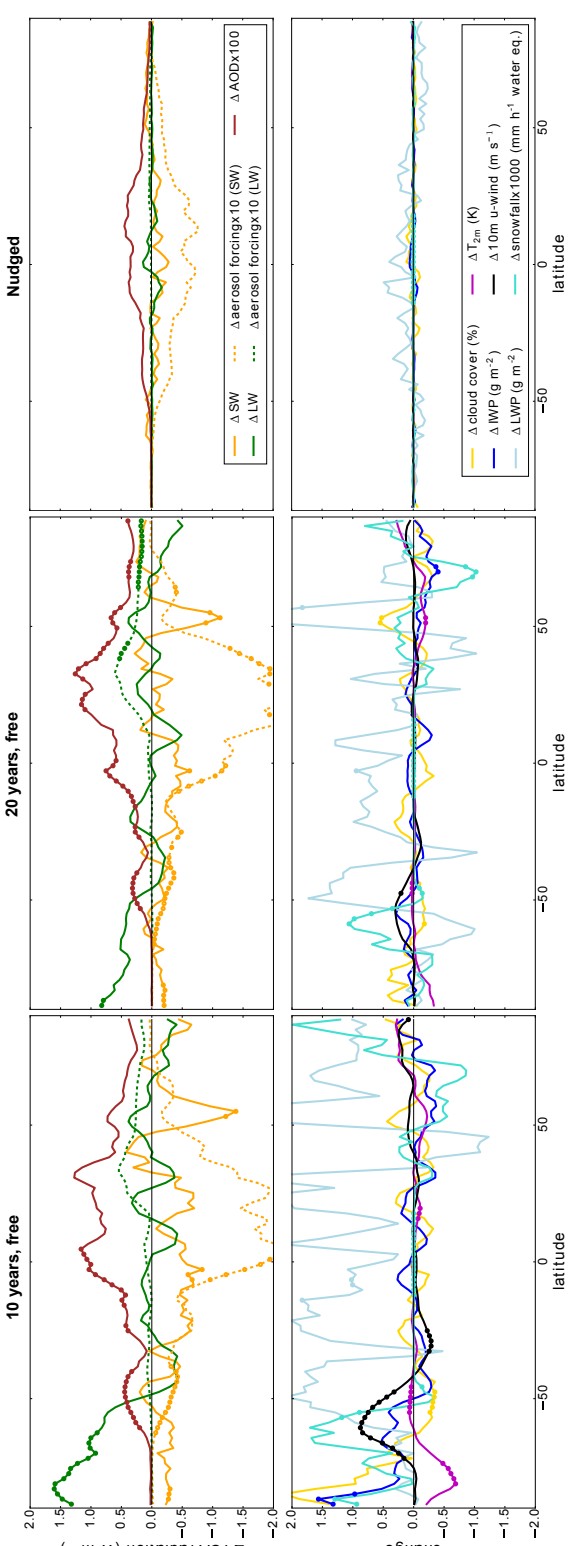

**Figure 12.** Top row: zonal, multi-year mean changes ("MOA100"-"noMOA") in the top-of-atmosphere (TOA) solar (SW) and terrestrial (LW) radiation in all-sky conditions, the corresponding changes in aerosol forcing at the TOA scaled up by an order of magnitude, and the change in aerosol optical depth (AOD) at the 550 nm wavelength scaled up by two orders of magnitude to fit the same scale on the y-axis. Bottom row: zonal, multi-year mean changes in cloud cover, ice water path (IWP), liquid water path (LWP), 2-metre temperature ($T_{2m}$), 10-metre zonal wind (10m u-wind), and water equivalent snowfall rate. The left panels are changes from the 10-year free simulations, the middle panels are changes from simulations identical to those on the left except they are extended for another 10 years, and those on the right are from the nudged simulations. Small circles on the curves indicate statistical significance at the 95 % level.





decrease with increasing temperature. When expressed in temperatures at which a $FF$ of 0.001 is reached, this is 7 °C lower for the Ickes et al. (2017) parametrisation compared to -21 °C for Niemand et al. (2012), assuming a spherical aerosol radius of 0.5 $\mu$m. The onset temperatures then further diverge with lower threshold $FF$s. The overall importance of MOA as an INP when compared to mineral dust is thus highly dependent on the choice of freezing parametrisations, for both MOA and dust.

This points to the need for more measurement data to better constrain the parametrisations, especially at warmer temperatures.

Extending the analysis one step further, impacts of MOA on clouds and climate are also investigated in this study. In general, weak to no statistically significant changes in cloud and climate variables are found due to the addition of MOA and due to MOA acting as an INP. More specifically, a decrease in in-cloud $r_{eff,i}$ by up to 3 to 9 % near the surface over both polar regions (statistically significant over the Southern Ocean) can be identified due to MOA initiating ice formation where previously only

supercooled droplets are present. In the climate variables, an overall statistically significant decrease in outgoing radiation at the TOA south of 50° S (associated with an increase in zonal winds over the Southern Ocean, cooling over Antarctica, and an increase in cloud ice properties over this region), is found in the analysis of the 10-year free simulations with the strongest MOA ice nucleation potential. The signal, however, is largely diminished when the same simulations are extended to 20 years. This points to the possibility that a ten-year mean is not sufficient to rule out internal variability in high latitudes as the reason

behind the observed signals, and has implications for future studies focusing on high latitude regions where longer simulations may be advised. When dynamical feedbacks are suppressed through nudged simulations, the changes are further diminished. We therefore conclude that any potential impact of the emitted MOA or MOA acting as an INP on the model climate is masked by the internal variability of the model. This can be partly attributed to the weak sensitivity of our model to heterogeneous ice nucleation (Ickes et al., in prep.), as well as to the weak ice activity of MOA when compared to dust.

**Appendix A: Offline comparison of OMF parametrisations**

Offline calculated monthly mean MOA concentrations at the two observational sites are shown in Fig. A1 using various OMF parametrisations. Monthly mean modelled 10-metre wind speeds and sea salt concentrations from a nudged simulation without MOA, averaged over the relevant period for each observational site (March 2002 to May 2009 for Mace Head and May 2003 to November 2007 for Amsterdam Island) are used in combination with the mean SeaWiFS observed chlorophyll concentrations

from the longer period. Two source regions are considered for each observational site: one following the region noted in the cited literature with the observational data, and the other approximated from Vergara-Temprado et al. (2017). OMFs are calculated offline for each source region using each OMF parametrisation and the chlorophyll concentrations and wind speeds from the corresponding region, as needed. As the OMF parametrisations are valid for the organic fraction during emission, the MOA concentration shortly after emission is approximated by taking the sea salt concentration in the lowest model level

with the derived OMF, following Eq. 1. The MOA concentration for the measurement site is then taken as the average of the concentrations over the entire source region. A schematic of the source regions and calculation method is shown in Fig. A2.




Notably, the calculated MOA concentration can vary by more than 0.1 $\mu$g m$^{-3}$ with slight shifts in the chosen source region, as can be observed by comparing solid and dotted curves in Fig. A1. When both source regions are considered, the Rinaldi et al. (2013) parametrisation is chosen as the best fitting to observations, though with a general slight underestimation.

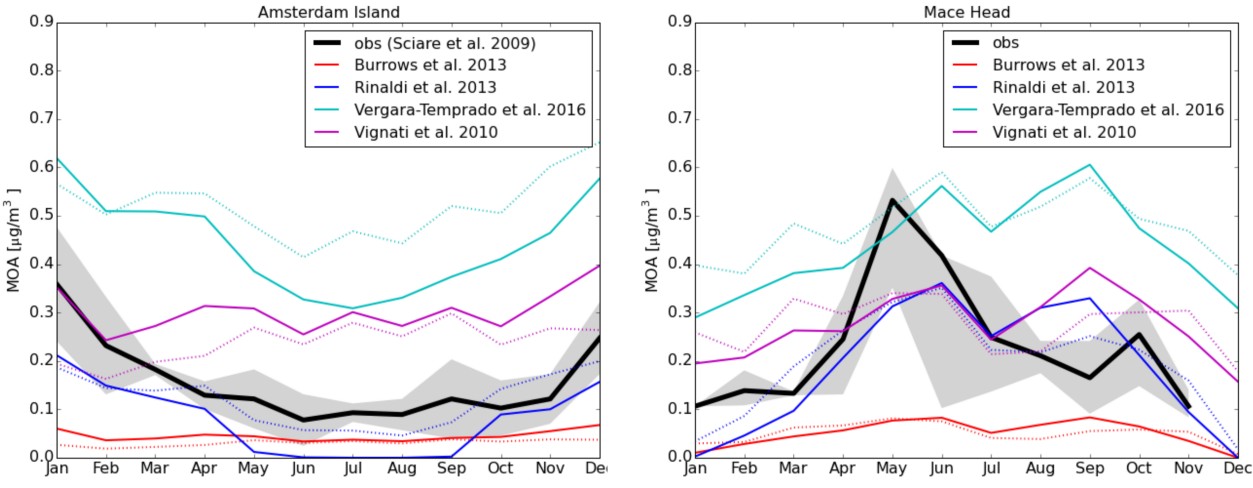

**Figure A1.** Comparison of MOA concentrations calculated offline using various OMF parametrisations. Coloured lines indicate offline calculated concentrations assuming the same source regions as the observational datasets; dotted lines of the same colours corresponds to the same parametrisations except for the use of source regions from Vergara-Temprado et al. (2017). The black lines and shaded areas are observational mean and the corresponding variances as described in Fig. 4.

## Appendix B: CMIP5 models with chlorophyll concentration output

5 Monthly mean of near-present day values from 2000 to 2005 of the historical ESM simulations are used for the "CMIP5chl" sensitivity study. The eight models for which such outputs can be obtained through the CMIP5 data portal, and thus used herein, are listed in Table A1.

*Author contributions.* The manuscript was written by W.T.K.H., who designed and performed the experiments, analyzed the outputs, and contributed to the implementation of MOA into the model. L.I. implemented the MOA tracers, ice nucleation due to MOA, as well as 10 the CNT parametrisation for dust following Ickes et al. (2017) into the model and contributed to discussions regarding the technical and scientific details of the study. I.T. implemented into ECHAM-HAM the sea salt and MOA emission parametrisations following Long et al. (2011) with an additional sea-surface temperature dependence described by Sofiev et al. (2011). M.R. and D.C. provided the observed MOA concentrations at Mace Head. U.L. provided scientific guidance in this study and oversaw the project. The manuscript has been read and approved by all co-authors.





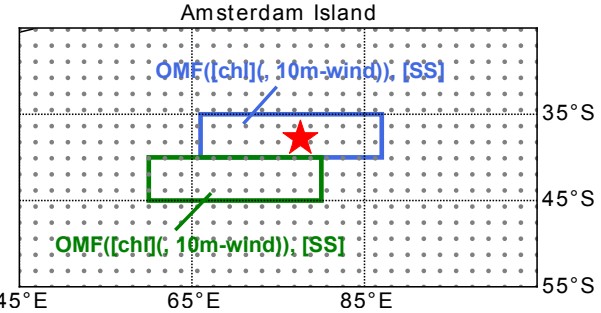

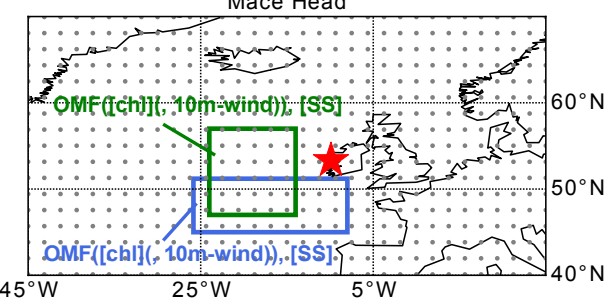

**Figure A2.** Source regions considered in offline calculations of MOA concentrations at each measurement site according to various OMF parametrisations. The area boxed in green is the source region from the relevant publication related to each observational dataset, while that in blue is approximated from Vergara-Temprado et al. (2017). The red star indicates the location of the measurement station. The gray dots that fill the space are model grid points. The words indicate that the offline calculations are done using the sea salt concentration from the lowest model layer ([SS]) and the OMF (which depends on the chlorophyll concentration, [chl], and in some cases also the 10 m wind speed) at each of the respective source regions.





**Table A1.** List of CMIP5 models containing sea surface chlorophyll concentration data used for the "CMIP5chl" simulation.

| Model name | Modelling centre or group | Mean of model versions, if multiple |
| --- | --- | --- |
| CanESM2 | Canadian Centre for Climate Modelling and Analysis | |
| CMCC-CESM | Centro Euro-Mediterraneo per I Cambiamenti Climatici | |
| CNRM-CM5 | Centre National de Recherches Météorologiques / Centre Européen de Recherche et Formation Avancée en Calcul Scientifique | |
| GISS-E2 | NASA Goddard Institute for Space Studies | GISS-E2-H-CC, GISS-E2-R-CC |
| HadGEM2 | Met Office Hadley Centre | HadGEM2-CC, HadGEM2-ES |
| IPSL-CM5 | Institut Pierre-Simon Laplace | IPSL-CM5A-LR, IPSL-CM5A-MR, IPSL-CM5B-LR |
| MPI-ESM | Max-Planck-Institut für Meteorologie | MPI-ESM-MR, MPI-ESM-LR |
| MRI-ESM1 | Meteorological Research Institute | |

*Acknowledgements.* The authors would like to acknowledge Jesús Vergara-Temprado for useful discussions regarding the emission and distribution of MOA in models. Thanks are also extended to Jurgita Ovadnevaite for discussions regarding AMS observations of marine organic matter at Mace Head. The research leading to these results has received funding from the European Union's Seventh Framework Programme (FP7/2007-2013) project BACCHUS under grant agreement n° 603445. For the CMIP5 data, we acknowledge the World Climate
5   Research Programme's Working Group on Coupled Modelling, which is responsible for CMIP, and the climate modeling groups (listed in Table A1 of this paper) for producing and making available their model output. The ECHAM-HAMMOZ model is developed by a consortium composed of ETH Zurich, Max Planck Institut für Meteorologie, Forschungszentrum Jülich, University of Oxford, the Finnish Meteorological Institute and the Leibniz Institute for Tropospheric Research, and managed by the Center for Climate Systems Modeling (C2SM) at ETH Zurich. Model simulations in this study were performed on the high performance computing cluster Euler, maintained by ETH Zurich's High-Performance Computing (HPC) group under the Scientific IT Services (SIS) section.





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
