# Peer review of "Global relevance of marine organic aerosol as ice nucleating particles"

_Atmospheric Chemistry and Physics, 2017_

## Referee Comment (RC1) · Anonymous Referee #1 · 22 Dec 2017

**Review of "Global relevance of marine organic aerosols as ice nucleating particles" by Huang et al.**

**General Comment:**

This study uses the ECHAM-HAM global climate model to examine the importance of the marine organic aerosol (MOA) in ice cloud formation when acting as ice nucleating particles. The authors evaluated different parametrizations in their climate model and found that the relative importance of MOA over mineral dust strongly depends on the used parametrizations. Additionally, it was found that the influence of MOA in the microphysical properties and radiative balance is mostly insignificant.

Given the large attention marine aerosol particles have recently gain due to the potential they have to catalyze the formation of ice clouds, and the large uncertainties coming from both observations and modelled predictions, it is necessary to conduct more research in this direction. This is an interesting study and it is a valuable addition to the literature. The paper is well written, and it can be accepted for its publication in ACP after the following minor and technical comments are incorporated in the final version. The reviewer has expertise in field and laboratory work and therefore, the following comments mostly focus on this part.

**Minor comments:**

1. Is ice multiplication considered in the ECHAM-HAM model? The present results indicate that MOA can nucleate ice particles at temperatures ranging from 0 to -10°C. This is the same temperature range at which the Hallett-Mossop mechanism is believed to take place. What is the influence of ice multiplication in the ICNC reported by the model, and if it is not considered what could be the uncertainty associated to this "omission"?
2. Tables 2, 3, and 4. Please add the meaning of the first column abbreviations in the table's caption. This will help the readers to follow the manuscript.
3. The reviewer have the impression that a deeper explanation on why high concentrations of MOA are not obtained in the Southern oceans when using the Rinaldi et al. (2013) emission.
4. P24 L10: Are additional uncertainties associated with this 10 additional years?
5. The grammar can be improved in several places.

**Technical Comments:**

1. The use of the word "aerosols" in the title and along the manuscript is incorrect. Aerosol refers to several particles, therefore it is not necessary to use the "s". Either use "Aerosol" or "Aerosol particles".
2. P1 L5-6, L18-19: I am wondering if the Wegener-Bergeron-Findeisen process is the only pathway for cloud glaciation. How about ice multiplication?

3. P1 L13: Please clarify if this statement refers to a global or regional scale.
4. P1 L17: Add a reference after "homogeneously".
5. P1 L18: Add a reference after "temperature".
6. P1 L20: Add more references after "precipitation".
7. P2 2: Add a reference after "effects".
8. P2 L4: I suggest to replace the Cziczo et al. (2017) reference with a more appropriate one (e.g., Kanji et al., 2017, Coluzza et al., 2017).
9. P2 L7: Why MOA is not as effective INP as mineral dust? This is based on who?
10. P2 L9: Replace "Marine organic aerosol" with "MOA".
11. P2 L9: Add a reference after "bursting".
12. P2 L11: Add a reference after "ocean".
13. P2 L12: Why insoluble organic matter only?
14. P2 L16: "heterogeneously frozen ice crystals" sounds a bit awkward.
15. P2 L18: Spell out ISCCP.
16. P2 L26: Add a reference after "robust".
17. P3 L22: Delete "cloud droplet number concentration" as this was already defined.
18. P4 L5: Why is sea salt independent of MOA?
19. P4 L10: A hygroscopicity parameter of zero is based on who?
20. P6 L7: Were the droplet freezing experiments conducted using a glass plate and a metal mesh? I don't think so. This sentence is confusing.
21. P15 Figure 4: Add "(left)" after "Island" and "(right)" after "Head" in figure caption.
22. P22 L7-8: How about ice multiplication? This also takes place at this warm temperatures in the presence of ice and supercooled drops.
23. P22 L17: Delete "(Wilks, 216)".
24. P25 L19-20: How about Burrows et al. (2013) and Yun and Penner (2013)?
25. P32 L12-13: Delete httP://www.atmos....
26. P32 L18: Correct the page numbers.

---

## Referee Comment (RC2) · S. M. Burrows (Referee) · 2 Feb 2018

Review of Huang et al. "Global relevance of marine organic aerosols as ice nucleating particles"

This paper addresses a question that is currently of some interest to the atmospheric ice nucleation community: the roles of marine organic aerosol (MOA), as contrasted with dust, in contributing to heterogeneous freezing in clouds and impacting cloud properties and the Earth's radiative budget.

Consistent with previous studies, the authors find that MOA frequently contributes the majority of mixed-phase cloud freezing nuclei at lower altitudes and high latitudes, especially over the Southern Ocean, with the freezing contribution from MOA greater

than that from dust on up to 40% of days under a variety of sensitivity cases.

In addition, the authors show that the relative importance of marine organic matter and dust as ice nuclei depends strongly on the choice of ice nucleation parameterizations for the two particle classes. This motivates a need for further experimental work to understand and quantify the factors that contribute to the observed variability in the INP activity of both dust and MOA, particularly at warmer temperatures.

Two published parameterizations of MOA INP (Wilson et al., 2015, drawn from sea surface microlayer samples, and DeMott et al., 2016, from laboratory-generated sea spray) are compared and shown to produce very different contributions of MOA to atmospheric freezing, although the percentage of time that MOA contribute the majority of INP is similar in each case.

Among other sensitivity experiments, the authors compare the impacts on cloud ice and liquid properties in simulations where MOA does not trigger freezing (noMOAfrz) and where it triggers freezing at a high rate (MOA100). The freezing induced by MOA is shown to produce a small (3%-9%), but statistically significant, decrease in zonal annual mean in-cloud mean ice crystal radius in near-surface clouds over the Southern Ocean. The changes in aerosols and aerosol-cloud effects are shown to result in statistically significant radiative cooling that is almost entirely attributable to an increase in outgoing longwave (terrestrial) radiation at high Southern latitudes, and that arises from a synergistic interaction of MOA acting as CCN and as scattering aerosol, and MOA acting as INP. MOA that do not act as INP, by contrast, are shown to produce a small zonal mean increase in reflected solar radiation, consistent with previous studies on the simulated effect of non-ice-nucleating MOA.

I have one comment that I feel needs to be seriously considered before publication, and which may require some corrections to one simulation. I am unclear on whether the DeMott et al. (2016) experimental data were interpreted and applied in the most appropriate manner. These data are best understood as characterizing the INP activity

of the total sea spray aerosol (including both salt and MOA), but a fit to these data appears to have been to only the MOA portion of the model's sea spray. If I have understood correctly what the authors did here, this one sensitivity case should be corrected before publication, as it will lead to a potentially significant underestimate of the globally-distributed concentrations of MOA INP that are implied by DeMott et al. (2016) and could create some confusion in the community about the appropriate interpretation of these data.

After the comments have been addressed, I strongly recommend this paper for publication. This study will be the most careful and comprehensive treatment of the global significance of MOA as INP to date. It constitutes an important contribution that advances the community's understanding of atmospheric INP sources, and of the importance of various uncertainties impacting the simulation of their contributions to freezing in clouds. In particular, it goes beyond most previous studies on the topic in several ways, in particular: (1) by comparing different observationally-derived representations of MOA INP activity for the first time in an atmospheric model, (2) by simultaneously considering the sensitivity to different parameterizations of sea spray emissions and of dust INP activity, and (3) by going beyond prediction of temperature-dependent INP concentrations to also investigate simulated impacts on clouds.

Major comments: 1. The most important comment I have is on the interpretation of the DeMott et al. (2016) dataset. DeMott et al. (2016) reported CFDC measurements of the ice nucleation activity of laboratory-generated sea spray aerosol particles that were composed of a combination of salt and organic matter. It's not completely clear to me whether the fit to the DeMott data (as presented in Fig 2) accounts for this in some way, but if I am understanding correctly, a fit was produced from the data in Fig 3 of DeMott et al. (2016), or a subset of those data, and then applied only to the surface area of the MOA portion of the sea spray aerosol - effectively, the surface area that the sea spray would have if the salt were removed (Eq. 4). It would be more appropriate to treat the DeMott data as representing the INP activity of the total sea spray surface

area of particles including both salt and MOA, and to then apply the fit to that total surface area as simulated by the model. This would produce an INP activity that is greater than what is calculated when only considering the organic portion of the sea spray, but still substantially less than what would be calculated using the Wilson et al. (2015) relationship. This needs to be clarified and discussed, and if am understanding correctly what was done, I think the calculations in this one sensitivity case need to be corrected in order to accurately represent the conclusions that can be drawn from the existing experimental data.

2. I'm aware that ECHAM-HAM cloud microphysics is continually being developed further and improved. To provide some perspective the simulated cloud impacts, it would be very helpful if the authors could comment a little further on the current level of confidence in the model's ability to credibly simulate mixed-phase cloud microphysics. In particular, the authors note that the model has a weak sensitivity to heterogeneous ice nucleation, but it's not completely clear what "weak" means here - i.e. is this in comparison to other models, to observational constraints, or to prior "expert judgement"? I recognize that this is the subject of another manuscript in preparation, but I think it is also important to contextualizing the results presented in this paper.

3. It would be helpful to see comparisons of the MOA100 versus noMOAfrz cloud properties on a seasonal basis, rather than only an annual mean basis. One would expect the strongest effects to occur in summertime at high latitudes (especially Southern hemisphere DJF), when longwave and shortwave radiation fluxes are greater. Could this explain why there are relatively large, and statistically significant effects on annual zonal mean LW and SW radiation at high Southern latitudes (Fig 11) despite limited effects of MOA INP on annual zonal mean cloud properties (Fig 9)?

Minor comments: 1. Figure 12: in the discussion of this figure, it is pointed out that using the statistical significance criterion applied here, some differences that are identified as statistically significant in the 10-year free-running simulation are no longer identified as significant in the 20-year free-running simulation, which the authors attribute to natural variability arising from dynamical feedbacks in the two simulations. How sensitive is this finding to the choice of significance threshold? To what extent does this finding hold if a stricter significance criterion is applied (e.g., $p < 1\%$ or $p < 0.1\%$ instead of $p < 5\%$)?

2. Appendix A: For the comparison with Mace Head and Amsterdam Island data, arguably it might be more appropriate to compare the model's MOA with observed water-insoluble organic carbon (as opposed to total organic carbon) – which would actually would produce a better agreement with the Rinaldi et al. (2013) parameterization, at least for the Amsterdam Island seasonal cycle. I don't think this matters for the conclusions of the paper, but it might be worth mentioning.

3. Appendix A: Rinaldi et al. (2013) is one of the better available chlorophyll-based MOA parameterizations and is clearly an appropriate choice for this study.

However, I think it would be appropriate to explicitly point out to non-specialist readers that considering the large differences between various models, it is not possible to infer from this figure that previous studies (Burrows et al., 2013; Vergara-Temprado et al., 2016) either significantly over- or underestimated MOA INP concentrations, compared to the current paper. This is already alluded to on p. 5, l. 3-5, but it should be pointed out again with respect to this specific analysis. Both previous studies documented that they produced OC concentrations consistent with the value typically observed in the remote marine atmosphere. In addition, the previous studies and Huang et al. have all pointed out that the uncertainty in modelling MOA INP concentrations is currently driven primarily by uncertainty regarding the INP activity of MOA, which is uncertain to at least two orders of magnitude. While important uncertainties remain in modelling of atmospheric MOA concentrations, these are comparatively well-constrained by observations of boundary-layer MOA concentrations from coastal stations and ships.

Typographical / technical comments: 1. P. 9, l. 19-20: "the total sea spray is split ... following Long et al. (2011)." I don't fully understand this sentence. Please reword for

clarity.

2. Maybe it's just my printer, but the hatching on some figures (e.g., Figure 9) is really difficult to see. Can this be improved?

3. P. 21, l. 5: "...further diagnostics are performed..." ("is" -> "are")
* * *

---

## Author Comment (AC1) · 16 May 2018

The authors would like to thank the reviewers for the valuable suggestions that helped to improve the paper. In particular, the suggestion to rerun the simulation for the DeMott et al. (2016) data led to discovery of two unit errors in the code concerning the freezing rate of MOA which subsequently propagated throughout the manuscript. Remedying the errors thus led to changes in the quantitative assessment of the results, though the qualitative conclusions remain largely unchanged.

To correct the unit errors, all simulations involving heterogeneous ice nucleation of MOA are re-run. This includes both nudged and free-running simulations, though the analyses performed for the former are mostly unaffected by the bug. The ice-activity

of MOA is now three and four orders of magnitude higher than previously stated, for the Wilson et al. (2015) parametrisation and following the fit to the DeMott et al. (2016) data, respectively. This puts them much closer to the ice activity of dust, though mostly still less ice-active. Subsequently, the simulation where the Wilson et al. (2015) parametrisation is scaled up by 100 (i.e. formerly "MOA100") is now deemed unnecessary. All figures in the manuscript (except Fig. 1 and those in the appendix, which are unaffected by the change) as well as numbers stated and interpretations thereof are now updated according to the new simulations. Figure 10 and related interpretations of it in the previous manuscript have also been changed completely as we found the new approach (looking at seasonal means) to be more helpful in our interpretations.

**Replies to reviewers:**

**Reviewer 1:**

Minor comments:

1. Is ice multiplication considered in the ECHAM-HAM model? The present results indicate that MOA can nucleate ice particles at temperatures ranging from 0 to -10°C. This is the same temperature range at which the Hallett-Mossop mechanism is believed to take place. What is the influence of ice multiplication in the ICNC reported by the model, and if it is not considered what could be the uncertainty associated to this "omission"?

—— Ice multiplication is not considered in the current simulations. Previous sensitivity tests have resulted in weak sensitivity of our model to such processes (not published). This is consistent with another manuscript in preparation by Ickes et al. where sedimenting ice crystals were found to dominate over other ice initiation processes in the mixed-phase regime. While our model is able to simulate the general observed cloud properties in the mixed phase regions (Lohmann and Neubauer 2018, ACPD), the

source of the ice crystals is not a constrained property, and indeed ice multiplication does have the potential to influence the results. This could be either an amplification of the changes in cloud properties due to heterogeneous ice nucleation (if ice is initiated where it is not previously present, subsequent ice multiplication can lead to an amplified signal in the cloud properties), or a dampening of the effects (if ice multiplication acts on pre-existing ice crystals from other sources, the influence by heterogeneous ice nucleation may be masked simply due to the lower rate of ice production). A note is added in Sect. 2.1 of the manuscript to point out the lack of consideration for such processes.

2. Tables 2, 3, and 4. Please add the meaning of the first column abbreviations in the table's caption. This will help the readers to follow the manuscript.

—— This has now been added in the captions.

3. The reviewer have the impression that a deeper explanation on why high concentrations of MOA are not obtained in the Southern oceans when using the Rinaldi et al. (2013) emission.

—— We now address this in slightly more detail in the manuscript: "...likely also due to the higher wind speeds and moderate chlorophyll concentrations (when using SeaWiFS maps) in these regions."

4. P24 L10: Are additional uncertainties associated with this 10 additional years?

—— The 10 additional years of simulation should not introduce any additional uncertainties. But rather, the purpose is to increase the sample size for better statistics.

5. The grammar can be improved in several places.

—— It's not immediately clear which parts of the manuscript the reviewer is referring to, but a few sentences have been re-worded and the grammar corrected in various places. Hopefully it helps with clarifications.

Technical Comments:

1. The use of the word "aerosols" in the title and along the manuscript is incorrect. Aerosol refers to several particles, therefore it is not necessary to use the "s". Either use "Aerosol" or "Aerosol particles".

—— Thank you for pointing this out. It has now been corrected.

2. P1 L5-6, L18-19: I am wondering if the Wegener-Bergeron-Findeisen process is the only pathway for cloud glaciation. How about ice multiplication?

—— Ice multiplication has now been noted in the manuscript as also a pathway to cloud glaciation.

3. P1 L13: Please clarify if this statement refers to a global or regional scale.

—— Global scale. This has now been added in the manuscript.

4. P1 L17: Add a reference after "homogeneously".

—— Added Rogers and Yau (1989).

5. P1 L18: Add a reference after "temperature".

—— Added Kanji et al. (2017).

6. P1 L20: Add more references after "precipitation".

—— Added Korolev (2007) in addition to Lohmann (2002).

7. P2 2: Add a reference after "effects"

—— Since this is a conclusion drawn from the previous (now cited) statements, reference was not added for this sentence.

8. P2 L4: I suggest to replace the Cziczo et al. (2017) reference with a more appropriate one (e.g., Kanji et al., 2017, Coluzza et al., 2017).

—— The Cziczo et al. (2017) reference is now changed to Kanji et al. (2017)

9. P2 L7: Why MOA is not as effective INP as mineral dust? This is based on who?

—— This statement is now changed to "While likely not as ice active as mineral dust especially in the colder mixed-phase temperatures, ..." and a reference is added for Vergara-Temprado et al. (2017).

10. P2 L9: Replace "Marine organic aerosol" with "MOA".

—— Done.

11. P2 L9: Add a reference after "bursting".

—— Added "e.g. Caroline and Keith, 2005".

12. P2 L11: Add a reference after "ocean".

—— Added "e.g. Bonsang et al., 1992"

13. P2 L12: Why insoluble organic matter only?

—— Only WIOM is considered because it is the most relevant type for ice nucleation, as was also done in previous studies (e.g. Burrows et al. 2013, Vergara-Temprado et al. 2017). Also, our model only contains a very simplistic parametrization for secondary organic aerosol formation (which is likely to be the main source of WSOM) and therefore is not suitable for examining the impact of these aerosol particles (more likely on the droplet activation).

14. P2 L16: "heterogeneously frozen ice crystals" sounds a bit awkward.

—— Now worded "heterogeneously formed ice crystals"

15. P2 L18: Spell out ISCCP.

—— Corrected.

16. P2 L26: Add a reference after "robust".

—— This sentence is a speculation (thus the words "may not be entirely") based on the argument laid out in the paragraph previously. There is no additional reference to support the claim. To perhaps express the idea better, the sentence is now reworded as: "…MOA may not be the sole missing ingredient responsible for the model shortfall."

17. P3 L22: Delete "cloud droplet number concentration" as this was already defined.

—— Replaced with simply "CDNC".

18. P4 L5: Why is sea salt independent of MOA?

—— This is a modelling choice in order to preserve the properties of the sea salt parametrization in the model, which was validated separately in previous studies. As MOA generally makes up a small fraction of the total sea spray, and as there is large uncertainties also in the sea salt emission, this assumption is not expected to have a strong impact on the results.

19. P4 L10: A hygroscopicity parameter of zero is based on who?

—— This was a rather arbitrary choice as no measurement data were available to constrain this property. Previously identified INP species (e.g. dust) indicate that a low hygroscopicity parameter would generally be expected for a good INP, and a value of zero was chosen for simplicity. The sentence is now changed slightly to "… for the last case, consistency with other potential INP candidates such as dust particles."

20. P6 L7: Were the droplet freezing experiments conducted using a glass plate and a metal mesh? I don't think so. This sentence is confusing.

—— The glass plate and metal mesh were referring to the collection method. The words have now been removed to avoid confusion.

21. P15 Figure 4: Add "(left)" after "Island" and "(right)" after "Head" in figure caption.

—— Added.

22. P22 L7-8: How about ice multiplication? This also takes place at this warm temperatures in the presence of ice and supercooled drops.

—— Due to changes in the manuscript associated with the model re-runs, the relevant paragraph is no longer included. In terms of interpretation of the general result, possible contributions from ice multiplication were not considered in the simulations.

23. P22 L17: Delete "(Wilks, 216)".

—— Corrected.

24. P25 L19-20: How about Burrows et al. (2013) and Yun and Penner (2013)?

—— These references have now been added.

25. P32 L12-13: Delete httP://www.atmos....

—— Done.

26. P32 L18: Correct the page numbers.

—— Is this referring to the pages 11-1 to 11-4 for Lohmann (2002)? This was double checked and does seem to be the correct page numbers.

**Reviewer 2: (Susannah Burrows)**

Major comments:

1. The most important comment I have is on the interpretation of the DeMott et al. (2016) dataset. DeMott et al. (2016) reported CFDC measurements of the ice nucleation activity of laboratory-generated sea spray aerosol particles that were composed of a combination of salt and organic matter. It's not completely clear to me whether the fit to the DeMott data (as presented in Fig 2) accounts for this in some way, but if I am understanding correctly, a fit was produced from the data in Fig 3 of DeMott et al. (2016), or a subset of those data, and then applied only to the surface area of the

MOA portion of the sea spray aerosol - effectively, the surface area that the sea spray would have if the salt were removed (Eq. 4). It would be more appropriate to treat the DeMott data as representing the INP activity of the total sea spray surface area of particles including both salt and MOA, and to then apply the fit to that total surface area as simulated by the model. This would produce an INP activity that is greater than what is calculated when only considering the organic portion of the sea spray, but still substantially less than what would be calculated using the Wilson et al. (2015) relationship. This needs to be clarified and discussed, and if am understanding correctly what was done, I think the calculations in this one sensitivity case need to be corrected in order to accurately represent the conclusions that can be drawn from the existing experimental data.

—— Thank you for pointing this out. This was indeed incorrectly represented in our model previously. The fit to DeMott et al. (2016) data has now been corrected to be applied for the total sea spray (MOA+SS). The simulation was re-run, and the manuscript corrected accordingly.

2. I'm aware that ECHAM-HAM cloud microphysics is continually being developed further and improved. To provide some perspective the simulated cloud impacts, it would be very helpful if the authors could comment a little further on the current level of confidence in the model's ability to credibly simulate mixed-phase cloud microphysics. In particular, the authors note that the model has a weak sensitivity to heterogeneous ice nucleation, but it's not completely clear what "weak" means here - i.e. is this in comparison to other models, to observational constraints, or to prior "expert judgement"? I recognize that this is the subject of another manuscript in preparation, but I think it is also important to contextualizing the results presented in this paper.

—— In the manuscript "What is triggering ice in mixed-phase clouds in ECHAM-HAM: the importance of ice nucleation" by Ickes et al. (in preparation) the role of ice nucleation for triggering ice in mixed-phase clouds is analyzed compared to other processes, such as sedimentation of ice crystals, vertical transport of ice crystals and entrainment

of ice crystals into a supercooled cloud. The analysis is based on the factorial method with the supercooled liquid fraction as a tracer for the microphysical structure of the mixed-phase clouds. It is shown that ice nucleation is less important compared to sedimentation of ice crystals, which is the dominant trigger process for ice in mixed-phase clouds in ECHAM-HAM. Heterogeneous ice nucleation and thus aerosol seems only to be an important for the initial ice in the supercooled cloud if there are no sedimenting ice crystals. This has implications for the sensitivity of the model to variations in ice nucleation parametrisations. However, the source of ice in mixed-phase clouds is in general not a well-constrained property in models and in this case does not have an impact on the general representation of the cloud microphysics in ECHAM-HAM in comparison to observations. A comparison of the mixed-phase cloud microphysics to observational data is shown in Lohmann and Neubauer (2018, ACPD), where the supercooled liquid fraction in ECHAM-HAM is shown to be relatively realistic when compared to Calipso observations.

We changed the wording in the manuscript at the end of the conclusions from "This can be partly attributed to the weak sensitivity of our model to heterogeneous ice nucleation (Ickes et al., in prep.), ..." to "This can be partly attributed to the weak sensitivity of our model to heterogeneous ice nucleation (due to the dominating contribution of cloud ice from sedimentation of crystals originating from cirrus levels; Ickes et al., in prep.), ...".

3. It would be helpful to see comparisons of the MOA100 versus noMOAfrz cloud properties on a seasonal basis, rather than only an annual mean basis. One would expect the strongest effects to occur in summertime at high latitudes (especially Southern hemisphere DJF), when longwave and shortwave radiation fluxes are greater. Could this explain why there are relatively large, and statistically significant effects on annual zonal mean LW and SW radiation at high Southern latitudes (Fig 11) despite limited effects of MOA INP on annual zonal mean cloud properties (Fig 9)?

—— Thank you for the suggestion. The seasonal mean comparisons for the cloud ice properties are now also included in the manuscript, and indeed we find the strongest

signal at high latitudes during summer. The signals previously identified in the LW and SW radiation at high southern latitudes, however, have become much weaker in the re-run simulations (pointing indeed to strong internal variability and possibly very strong anomalous years in the previous simulations) and the propagation of changes in cloud properties to changes in the mean TOA radiative properties still appears to be weak.

Minor comments:

1. Figure 12: in the discussion of this figure, it is pointed out that using the statistical significance criterion applied here, some differences that are identified as statistically significant in the 10-year free-running simulation are no longer identified as significant in the 20-year free-running simulation, which the authors attribute to natural variability arising from dynamical feedbacks in the two simulations. How sensitive is this finding to the choice of significance threshold? To what extent does this finding hold if a stricter significance criterion is applied (e.g., $p < 1\%$ or $p < 0.1\%$ instead of $p < 5\%$)?

—— In the re-run simulations, the difference between the 10-year and 20-year free-running simulations is much weaker. A stricter $p < 1\%$ is now used in the manuscript, and the reduction in significance of the changes can still be noted (though on a smaller scale). On the contrary, when a more lenient $p < 5\%$ is used, the latitudes with significant changes remained more consistent between the two analysis periods.

2. Appendix A: For the comparison with Mace Head and Amsterdam Island data, arguably it might be more appropriate to compare the model's MOA with observed water-insoluble organic carbon (as opposed to total organic carbon) – which would actually would produce a better agreement with the Rinaldi et al. (2013) parameterization, at least for the Amsterdam Island seasonal cycle. I don't think this matters for the conclusions of the paper, but it might be worth mentioning.

—— The comparison is actually indeed with the observed water-insoluble organic carbon, which is then converted to WIOM to be consistent with the modelled MOA. This is now also stated in the manuscript to hopefully clear up any ambiguity.

3. Appendix A: Rinaldi et al. (2013) is one of the better available chlorophyll-based MOA parameterizations and is clearly an appropriate choice for this study. However, I think it would be appropriate to explicitly point out to non-specialist readers that considering the large differences between various models, it is not possible to infer from this figure that previous studies (Burrows et al., 2013; Vergara-Temprado et al., 2016) either significantly over- or underestimated MOA INP concentrations, compared to the current paper. This is already alluded to on p. 5, l. 3-5, but it should be pointed out again with respect to this specific analysis. Both previous studies documented that they produced OC concentrations consistent with the value typically observed in the remote marine atmosphere. In addition, the previous studies and Huang et al. have all pointed out that the uncertainty in modelling MOA INP concentrations is currently driven primarily by uncertainty regarding the INP activity of MOA, which is uncertain to at least two orders of magnitude. While important uncertainties remain in modelling of atmospheric MOA concentrations, these are comparatively well-constrained by observations of boundary-layer MOA concentrations from coastal stations and ships.

—— This point is now explicitly noted at the end of the paragraph in Appendix A: "It should be noted, however, that the assessment of the wellness of fit to observations is highly model-dependent. Thus while suitable for choosing an appropriate OMF parametrization for this particular model, no generalizations should be drawn with regards to the quality of the individual parametrisations. Indeed, each of the OMF parametrisations have been separately validated in their respective studies and found to fit well to observations under their respective set-ups."

Typographical / technical comments:

1. P. 9, l. 19-20: "the total sea spray is split . . . following Long et al. (2011)." I don't fully understand this sentence. Please reword for clarity.

—— Now reworded as: "…when both MOA and SS are emitted following Long et al. (2011), the total sea spray is divided between sea salt and MOA components. This

would therefore also lead to decreases in the sea salt emission rates when compared to the control simulation."

2. Maybe it's just my printer, but the hatching on some figures (e.g., Figure 9) is really difficult to see. Can this be improved?

–––– Yes. This was due to the plotting program used, and have now been changed to hopefully improve the clarity of the hatching.

3. P. 21, l. 5: ". . .further diagnostics are performed. . ." ("is" -> "are")

–––– Due to changes in the results associated with the reruns, the relevant sentence is no longer included in the manuscript.